# Look Around and Refer: 2D Synthetic Semantics Knowledge Distillation for 3D Visual Grounding

**Eslam Mohamed Bakr, Yasmeen Alsaedy, Mohamed Elhoseiny**
King Abdullah University of Science and Technology (KAUST)
{eslam.abdelrahman, yasmeen.alsaedi, mohamed.elhoseiny}@kaust.edu.sa

## Abstract

The 3D visual grounding task has been explored with visual and language streams comprehending referential language to identify target objects in 3D scenes. However, most existing methods devote the visual stream to capturing the 3D visual clues using off-the-shelf point clouds encoders. The main question we address in this paper is *"can we consolidate the 3D visual stream by 2D clues synthesized from point clouds and efficiently utilize them in training and testing?"*. The main idea is to assist the 3D encoder by incorporating rich 2D object representations without requiring extra 2D inputs. To this end, we leverage 2D clues, synthetically generated from 3D point clouds, and empirically show their aptitude to boost the quality of the learned visual representations. We validate our approach through comprehensive experiments on Nr3D, Sr3D, and ScanRefer datasets and show consistent performance gains compared to existing methods. Our proposed module, dubbed as Look Around and Refer (LAR), significantly outperforms the state-of-the-art 3D visual grounding techniques on three benchmarks, i.e., Nr3D, Sr3D, and ScanRefer. The code is available at https://eslambakr.github.io/LAR.github.io/.

## 1 Introduction

Visiolinguistic modeling has attracted much attention over the past decade due to impact on various areas, including visual question answering [3, 36], 3D scene captioning [7], image-text matching [26, 44], visual relation detection [25], and scene graph generation [42]. Consequently, understanding natural language utterances and grounding them to real-world scenarios is attractive due to their importance in many real applications, such as robot navigation and interaction in indoor environment [22] [40].

Visual Grounding (VG) has been explored in recent years in the context of 2D [16, 30, 46, 55, 57], extending object detection to a more complex task. However, for real robots to intelligently ground a given utterance on a natural scene in a challenging environment like a crowded indoor or outdoor scene, 3D representations are needed to understand the surrounding environment better. Therefore, 3D visual grounding task have been proposed by ReferIt3D [2] and Scan-Refer [5]. The ReferIt3D [2] introduces two datasets containing natural and synthetic language utterances,

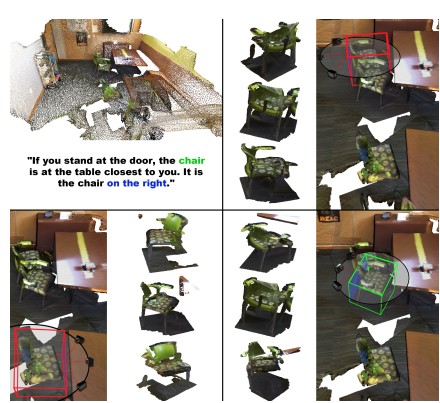

Figure 1: Overview of our proposed approach, where we aim to localize a referred object given an input utterance and 3D scene. Our novel approach generates 2D synthetic images for each object in the scene by placing virtual cameras surrounding the object. This novel approach enables us to leverage the rich 2D semantics representation without needing authentic extra 2D images not to limit potential application scenarios.

36th Conference on Neural Information Processing Systems (NeurIPS 2022).

Table 1: Comparison of various 3D visual grounding models. Where GT indicates ground truth boxes are used, Pred. indicates a pretrained detector is used to get the objects proposals. We show that Looking-outside [18] and SAT [52] are using additional inputs such as the over all scene point clouds $S_{pc}$, and extra 2D images $2D_{Img}$, respectively. $O_{pc}$ refers to object's point cloud.

| | Refit3D [2] | Lang. Refer [37] | InstRef [58] | TGNN [17] | Looking outside [18] | D3Net [6] | TransRef3D [13] | 3DVG [59] | SAT [52] | Ours |
|---|---|---|---|---|---|---|---|---|---|---|
| Obj. Proposal | GT | GT | Pred. | Pred. | Pred. | Pred. | GT | Pred. | GT | GT |
| Lang. Encoder | RNN | DistilBert | GloVE | GloVE | RoBERTa | GRU | DistilBert | GloVE | BERT | BERT |
| Visual Input | $O_{PC}$ | $O_{PC}$ | $O_{PC}+S_{PC}$ | $O_{PC}$ | $O_{PC}+S_{PC}$ | $O_{PC}$ | $O_{PC}$ | $O_{PC}$ | $O_{PC}+2D_{Img}$ | $O_{PC}$ |

namely Nr3D and Sr3D, respectively, which aims to determine the referent object from pre-defined objects set for each scene. In contrast, ScanRefer [5] aims to predict 3D bounding boxes for the referent object based on a given language description and 3D point clouds. Both ScanRefer [5] and ReferIt3D [2] are built on the ScanNet dataset [8]. The critical difference, which made ReferIt3D [2] more challenging, is that in each scene, several object instances belong to the same fine-grained category as the referent, namely distractors.

Due to the sparsity and disorganization of the 3D point clouds, SAT [52] explores utilizing the 2D images provided by the ScanNet dataset [8] during training to assist point-cloud-language joint representation learning. SAT [52] proposes three variants; 1) The baseline, non-SAT, relies only on the 3D point clouds while tackling the grounding task. 2) Utilizing 2D semantics during the training phase only and masking them while inference, we termed it SAT. 3) Utilizing the 2D semantics in both phases, i.e., training and testing, we termed it SAT-GT. However, requiring extra 2D inputs in training or inference limits potential application scenarios. To tackle this drawback, we hypothesized that incorporating synthetic 2D semantics will consolidate the 3D learned representation while not requiring any additional info. To this end, we propose a novel 2D Synthetic Images Generator (SIG) that projects the 3D point clouds into multi-view 2D images, as shown in Figure 1. To the best of our knowledge, LAR is the first method that augment learning of 3D tasks with synthetic 2D semantics. In addition, we propose a novel multi-modal transformer-based architecture. Our proposed architecture, LAR, on Nr3D outperforms SAT [52] by boosting the accuracy by 1.6% while exceeding all existing approaches that did not use any additional information than the 3D point clouds by a wider margin, i.e., 5.0% to the closest one. Also, LAR architecture achieves state-of-the-art results on Sr3D and ScanRefer benchmarks, 59.35% and 54.6%, respectively. Another variant is tailored from our architecture to be compared against SAT-GT, where geometry information and ground-truth class labels are concatenated to the 2D images; synthetic in our case and original images in SAT case. Our LAR architecture outperforms SAT-GT by 11.7% ( 62% for our method and 50.3% for that). Our contributions are summarized as follows:

- We propose a novel End-to-End multi-modal transformer-based architecture, LAR, for the 3D visual grounding task that relies only on 3D point clouds, which exploits the synthetic 2D images generated by our SIG module.
- We verify the effectiveness of our LAR architecture throughout extensive experiments with various benchmarks, i.e., Nr3D, Sr3D, and ScanRefer, which show that utilizing our 2D synthetic semantics significantly enhances the results.
- Through detailed analysis, we examine our method's internal behavior and validity.

## 2 Related Work

Our novel architecture draws success from several areas, including 2D visual grounding, 3D visual grounding, and 2D semantics in 3D tasks.

**–2D visual grounding.** 2D visual grounding is locating an object based on a natural language description. Flickr30k [32] and ReferItGame [20] generated referring expressions for a particular object in an image used to identify which object is being referred to. Prior works [56, 27, 29] have attempted to localize these referred objects based on the given expressions by focusing on the bounding-box level. In contrast, [53, 10] focus on the pixel- level predictions. Another 2D visual grounding taxonomy is the one-stage, also known as a single-stage framework, versus the two-stage framework. Single-stage methods (e.g., [38, 49, 50]) fuse the linguistic query with each pixel/patch of the image, and generate a set of possible bounding box candidates. In contrast, the two-stage framework [45, 54, 56] uses the visual image semantics to generate object proposals. Afterward, the

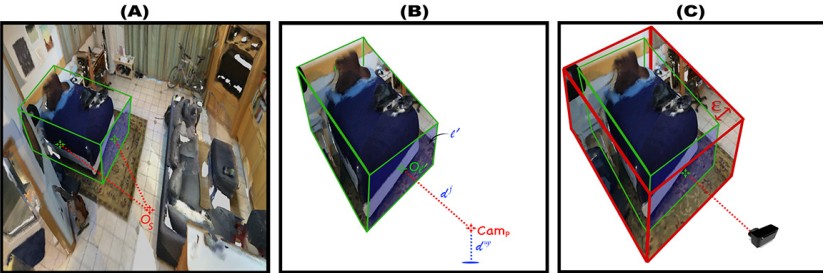

Figure 2: Simplified overview of our 2D Synthetic Images Generator (SIG) module. First, we determine the prominent face of each object w.r.t the scene center. Then, the camera is located at a distance $d^f$ from that face and at a distance $d^{up}$ from the room's floor. Finally, we randomly extend the region of interest by $\epsilon$.

grounding prediction is selected by comparing each proposal to the language query. Consequently, 2D visual grounding methods can guide learning 3D relationships. Driven by this, we explore using 2D synthetic semantics to consolidate the 3D visual stream.

–**3D visual grounding.** Table 1 summarizes the existing 3D visual grounding (3D VG) models whether a ground truth object proposal is used or a pre-trained detector is utilized to generate the proposals. Also, it summarizes whether extra info is used alongside the object's 3D point cloud or not. 3D visual grounding field has undergone extensive research and made considerable progress over the last few years [2, 5, 12, 17, 59, 1, 37, 13, 58, 19, 52, 6, 19]. A 3D visual grounding task uses language instruction along with a 3D scene [8] as input in order to estimate a bounding box that allocates the most relevant object to the provided expression. However, the literature offers several extensions, such as using auxiliary input to improve localizing 3D objects [52]. Previous work [2, 12, 17, 59, 1, 13, 58, 19], focuses on improving the language and 3D scene fusion by adopting a two-stage framework. The first stage outputs 3D object proposals given a 3D point cloud representing a scene. Secondly, visual and linguistic features are fused to compute a confidence score for each 3D object proposal. The first stage could be skipped by directly using the ground-truth proposals instead of predicting them [52] [13] [2]. In contrast, [6, 19, 59] do not assume the access to the ground-truth objects and use an object detector to extract the proposals. Intuitively, if the detector fails, the referring performance will significantly decrease. Therefore, we focus on the main training objective related to the grounding task, which is learning 3D visual-language joint representations by assuming access to the ground-truth boxes. In addition, we explore the other setting that utilizes detector-generated proposals, which is mentioned in the Appendix.

–**2D semantics in 3D tasks.** 2D semantics contains useful information that can be used to detect 3D objects. The representations of visual scenes consist of point clouds, meshes, or volumes, while 2D images consist of pixel grids. Combining the best of two worlds can enhance 3D object localization. Researchers have attempted to solve the 3D detection problem by adopting different ways of representing RGB-D data [34, 48, 23, 15, 41, 24]. By either projecting the 2D image on the 3D scene [34, 48, 24] or fuse both 2D features with the 3D [23, 15, 43, 41], [33]. SAT [52] leveraged 2D semantics to perform the 3D VG task. However, requiring extra 2D inputs limits potential application scenarios. In contrast, in this work, we generate synthetic 2D images during the training phase, which is also used in the inference phase. The synthetic images enable us to consolidate the 3D visual stream by 2D semantics without requiring extra 2D inputs.

## 3   Look Around and Refer (LAR)

LAR aims to enhance the visual module, which in turn enhances the final objective,i.e., referential accuracy. By incorporating the synthetic 2D semantics alongside the 3D point clouds, LAR inherently captures the correlations between the 2D and the 3D streams. To design a general framework to fit in any 3D application, when the input is 3D point clouds, the need for extra information, such as 2D semantics, should be bypassed. To this end, a 2D synthetic images generator, namely SIG, is introduced, which is discussed in Section 3.1. Then, we demonstrate how to learn robust representations by aligning the generated 2D semantics with the 3D ones by incorporating our SIG module into our proposed architecture, depicted in Section 3.2.

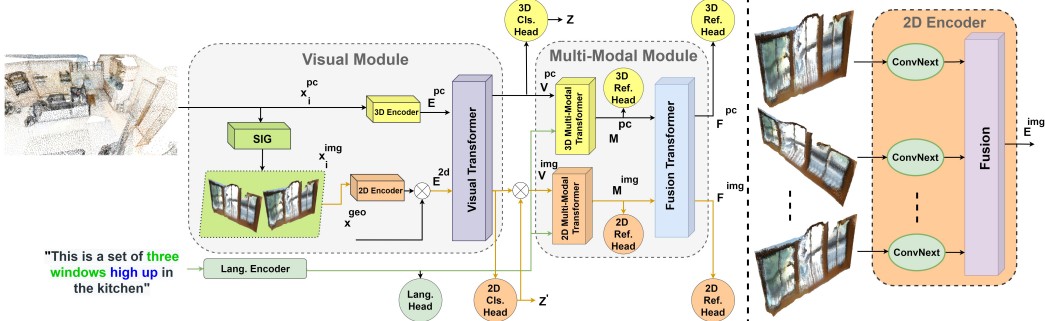

Figure 3: Detailed overview of LAR. The Visual Module incorporates rich 3D representation from the extracted 3D object points with the 2D synthetic image features using Visual Transformer. The 2D synthetic images are first extracted by SIG and then processed by a shared ConvNext backbone. Simultaneously, language descriptions are transformed into tokens and embedded into a feature vector. Then, the Multi-Modal Module fuses the output of the Visual Module separately by two Transformers. Finally, the output of the Multi-Modal Transformers is processed by Fusion Transformers.

## 3.1 Look Around: 2D Synthetic Images Generator

LAR exploits synthetic 2D semantics to reinforce the learned 3D representations. Consequently, this section will demonstrate our 2D Synthetic Images Generator (SIG) module that projects arbitrary 3D Point clouds representing an object to a 2D image. First, we introduce the essential operations of our module on a single image, then show how we extend it to fulfill a multi-view setup.

A 3D scene, $S \in \mathbb{R}^{N \times 6}$, is represented by $N$ points with spatial and color information, i.e., XYZ, and RGB, respectively. Following the previous works [2] [1] [37] [52], the object proposals $x_i^{pc}$ for each scene are known, either generated by one of the off-the-shelf 3D object detectors or manually annotated. The object proposal $x_i^{pc} \in \mathbb{R}^{N' \times 6}$ is a subset of the whole scene, where $N' < N$ and $N'$ represents the number of points that represent the object $i$.

Given a scene $S$ and an object proposal $x_i^{pc}$, our SIG module generates a 2D synthetic image $x_i^{img}$ represents the object $i$. The main idea is to place a virtual camera in front of the fed object proposal, then project its points $x_i^{pc}$ into the 2D plane to generate the corresponding 2D image $x_i^{img}$. The camera's location should be set carefully to generate a decent image representation for an object $i$. As shown in part (A) in Figure 2, after excluding the upper and lower faces from the fed object proposal, we get the center of the four remaining faces, then calculate the distance between each face center $O_i^l$, and the center of the scene $O_s$, where $l$ ranging from 0 to 3. The nearest face $l'$ from the scene center $O_s$ is the prominent one, colored with blue in Figure 2, which guarantees that we look at the object $i$ from inside the room. The camera is located at a distance $d^f$ from the prominent face $l'$ and at a distance $d^{up}$ from the room's floor; thus, the camera position is interpreted as $Cam_p = [d_x^f, d_y^f, d^{up}]$, as shown in part (B) in Figure 2. The orientation of the camera frame is defined as follows. The z-axis is aligned with the principal axis of the camera, and two other orthogonal directions (x, y) align with the corresponding axes of the image plane. Failure cases are shown in Figure 5 when the camera is located outside the room.

To project the points $x_i^{pc}$, representing object $i$, from the scene coordinates to the camera coordinates, a transformation matrix $M \in \mathbb{R}^{4 \times 4}$ needs to be defined. Given the camera position $Cam_p$ and the center of the prominent face $O_{l'}$, the transformation matrix $M$ can be interpreted as, $\begin{bmatrix} V^r & V^{up} & V^f & Cam_p \\ 0 & 0 & 0 & 1 \end{bmatrix}$ Where the camera direction is defined by three vectors; $V^f$, $V^r$, and $V^{up}$. We define our virtual camera using the Unified Camera Model (UCM), which has five parameters $I = [\gamma_x, \gamma_y, c_x, c_y, \zeta]$ Following the UCM, projection is defined as follows: $\pi(x_i^{pc}, I) = \begin{bmatrix} \gamma_x \frac{x}{\zeta d + z} \\ \gamma_y \frac{y}{\zeta d + z} \end{bmatrix} + \begin{bmatrix} c_x \\ c_y \end{bmatrix}$,

Afterward, the projected 3D point clouds are assigned to a 2D grid map. To mitigate overfitting, we randomly extend the region of interest by $\epsilon$ as demonstrated in part (C) in Figure 2.

Point clouds differ from 2D images, where every pixel contains spatial and texture details. Moreover, the object is represented in a view-invariant way, unlike the images that are view-dependent. To mitigate this drawback in the image representation, a multi-view setup is constructed as shown in Figure 1 where $v$ cameras are mounted in a circle around the prominent face $l'$. The pseudo-code for our SIG module is attached in the supplemental materials.

### 3.2 Refer: Visiolinguistic Transformer Architecture

We present our approach, depicted in Figure 3, for learning multi-modal representations of a 3D scene associated with language instruction. First, the proposed visual module enriches the 3D representations by utilizing the 2D synthetic images generator, namely the SIG module, described in Section 3.1. Thanks to it, we do not require access to any imagery sensor during training or testing. Then, the multi-modal module learns joint representations targeted to the final task, i.e., referring to an object in the scene based on the language instruction.

–**Visual Module.** Given a 3D scene $S \in \mathbb{R}^{N \times 6}$ combined with agnostic multiple 3D object proposals $x_i^{pc} \in \mathbb{R}^{N' \times 6}$, we aim to classify each object proposal by predicting the semantic labels $z$. First, for each object $i$ in the scene $S$, we generate multi-view synthetic 2D images $x_i^{img} \in \mathbb{R}^{v \times H \times W \times 3}$ from the 3D object's point cloud $x_i^{pc}$, where $v$ is the number of views for each object, $H$ and $W$ are the spatial dimensions for the generated images. Second, we employ PointNet++ [35] as a 3D encoder, producing 3D features $E_i^{pc} \in \mathbb{R}^{1 \times dim_{pc}}$. Hence, the Tiny-ConvNext [28] is adopted as a 2D encoder, that produce $y_i^{img} \in \mathbb{R}^{v \times H' \times W' \times dim_{img}}$. Where $H'$ and $W'$ represent a coarse spatial grid of spatial features, each represented by a $dim_{img}$. Then, the spatial dimensions are squeezed producing $o_i^{img} \in \mathbb{R}^{v \times dim_{img}}$. Since our 2D encoder encodes multiple views $v$ for the same object $i$, a fusing mechanism is needed to fuse the views' features. Various possibilities are explored to settle down on the best fusion mechanism. We can interpret this as a fully connected layer to fully capture the views and cross-channel interactions. In contrast, to learn the views' interactions and neglect the cross-channel relations, a 1D-Conv. could be utilized to avoid involving tremendous parameters as in the fully connected case. Using the 1D-Conv. with $k_{size} = v$ as a fusion layer produce $E_i^{img} \in \mathbb{R}^{1 \times dim_{img}}$, where $k_{size}$ is the kernel size. The fused 2D features $E_i^{img}$ and the 3D features $E_i^{pc}$ are concatenated and fed to a visual transformer, which is composed of several stacked transformer layers. The visual transformer is utilized to capture the cross-correlation between the 2D, and the 3D features outputting refined features, $V^{pc}$ and $V^{img}$, guided by three weighted auxiliary losses, i.e., alignment loss between $V^{pc}$ and $V^{img}$, and two visual classification losses.

–**2D semantics.** Inspired by SAT [52], we explore different 2D semantic information types combinations. Where we formulate our 2D semantics as follows:

$$E_i^{2d} = LN(\phi(E_i^{img}, LN(W_1 x_i^{geo}), z_i')), \tag{1}$$

where $z_i'$ is the predicted class labels, $W_1$ are learned projection matrices, LN is layer normalization, and $\phi$ indicates concatenation. We encode each object individually using different semantic information types, i.e., visual semantics $E_i^{img}$, and geometry semantics $x_i^{geo} \in \mathbb{R}^{1 \times 30}$. The geometry semantics, $x_i^{geo}$ represents the bounding box coordinates of object $i$ in the scene space and encode the virtual camera parameters, i.e., intrinsic and extrinsic parameters, used while generating the 2D image $x_i^{img}$. Thus, it can be interpreted as a positional encoding for the vision transformer, where the interaction among objects is captured. In contrast, SAT [52] utilizes a ground-truth 2D semantics, where the $z_i'$ denotes the manually annotated class labels, and the $E_i^{img}$ represents the embedding generated from real 2D images captured by imagery sensor [8].

–**Multi-Modal Module.** Using a pre-trained BERT model [11] followed by a language classification head adopted from Referit3D [2], we embed words query $Q$. After embedding each of the visual features using the aforementioned visual module, we fuse the input modalities $Q$, $V^{pc}$, $V^{img}$ simultaneously by using two different transformers; 3D Multi-Modal Transformer and 2D Multi-Modal Transformer, as shown in Figure 3. In 3D Multi-Modal Transformer, we jointly encode the two input modalities $V^{pc}$ and $Q$ to understand better the referring expression that captures co-occurrence relationships between different objects in the scene. Meanwhile, the same words query $Q$ are encoded with $V^{img}$ to enhance context-aware representation. A separate 2D and 3D referring heads are adopted from Referit3D, which are auxiliary tasks. Finally, the output of the multi-modal transformers is processed by a Fusion Transformer that chooses the referred object from the fused features, $F^{pc}$ and $F^{img}$.

### 3.3 Losses

Our architecture, termed LAR, consists of two visual streams, i.e., 3D and 2D, in addition to one language stream. To refer to the object correctly guided by the input utterance, we must first classify the objects accurately. Thus, two auxiliary losses are employed to optimize the visual module; 3D classification loss $\mathcal{L}_{cls}^{3D}$ and 2D classification loss $\mathcal{L}_{cls}^{2D}$. In addition, We adopted Referit3D [2] auxiliary loss for language classification loss $\mathcal{L}_{cls}^{lang}$ to cooperate in the visual grounding task by recognizing the referred object class according to the description. Hence, two auxiliary grounding losses ($\mathcal{L}_{Eref}^{3D}$ and $\mathcal{L}_{Eref}^{2D}$, termed as early referring losses, are exploited before the fusion transformer. As a result of these early referring losses, the model learns the correlation between language description and the visual features separately, which enhances the performance. Additionally, we adopted object correspondence loss $\mathcal{L}_{cor}$ from SAT to encourage the 3D model to distill the knowledge from the 2D stream. Finally, two grounding losses, i.e., $\mathcal{L}_{ref}^{3D}$ and $\mathcal{L}_{ref}^{2D}$ were added after the fusion transformer to predict the scores of each proposal. We optimize the whole model in an End-to-End manner, unlike SAT which learns the 2D features $E_i^{img}$ separately, with the following loss function:

$$\mathcal{L} = \lambda_{cls}*(\mathcal{L}_{cls}^{3D}+\mathcal{L}_{cls}^{2D})+\lambda_{Eref}*(\mathcal{L}_{Eref}^{3D}+\mathcal{L}_{Eref}^{2D})+\lambda_{ref}*(\mathcal{L}_{ref}^{3D}+\mathcal{L}_{ref}^{2D})+\lambda_{cor}*\mathcal{L}_{cor}+\lambda_{lang}*\mathcal{L}_{cls}^{lang}, \quad (2)$$

where $\lambda_{cls}$ is the object classification loss weight, $\lambda_{Eref}$ is the early referring loss weight, $\lambda_{ref}$ is the final referring loss weight, $\lambda_{cor}$ is the object correspondence loss weight, and $\lambda_{lang}$ is the language classification loss weight. Our implementation details are detailed in the Appendix.

## 4 Experiments

### 4.1 Datasets

We evaluate our proposed 3D visual grounding approach, LAR, on three benchmarking datasets, Nr3D [2], Sr3D [2], and ScanRefer [5]. Nr3D, 3D Natural reference dataset [2], consists of 41.5K natural, free-form utterances collected by humans using a referring game between two humans. As opposed to Sr3D [2], which consists of 83,5K synthetic utterances. Consequently, ScanRefer provides 51.5K utterances of 11K objects for 800 3D indoor scenes.

### 4.2 Evaluation Metrics

we followed the same metrics followed by the previous work [2] [52], where three evaluation metrics are mainly used, i.e., the referring accuracy, visual classification accuracy, and language classification accuracy. In the case of Nr3d and Sr3d datasets, we followed the previous work convention, assuming the proposals are generated with GT objects. Following this assumption, the 3D grounding problem is reformulated as a classification problem, where the model objective is to classify the referred object among the other objects correctly, termed referring accuracy. The referring accuracy is calculated based on whether the model picks the correct proposal from a set of X proposals. Alongside the main objective, which is the referring accuracy, two auxiliary metrics are defined to evaluate the performance of each stream individually, i.e., the visual classification accuracy and the language classification accuracy. Given agnostic GT-proposals, the visual classification accuracy is measured based on whether the model predicts the object's class correctly or not from a predefined set of possible classes. Similarly, language classification accuracy is another auxiliary metric that assesses the performance of the linguistic branch. This branch aims to correctly predict the class of the referred object based on the input utterance.

### 4.3 Implementation Details

–**Input Configuration.** For the 3D input, we randomly sample 1024 points per proposal from its point cloud segment. Furthermore, for the synthetic 2D input, we follow the algorithm, which is scrutinized in Section 3.1. During the testing phase, the camera is set 2 meters away from the prominent face $l'$, i.e., $d^f = 2$, and 1 meter away from the floor, i.e., $d^{up} = 1$. In addition, the region of interest extension ratio $\epsilon$, depicted in Figure 2 part (C), is set to 2%. The generated image is a square with a size of 32 pixels. Five views are generated per object, where the angle $\theta$ between each camera is $30°$, demonstrated in Figure 1. Thus the camera is set to the following angles ( $0°$,$30°$, $-30°$,$60°$,$-60°$ ).

Table 2: Ablation Study of 2D synthetic image resolution and multi-view setup on Nr3D [2].

| Image Size | Multi-Views | Encoder | Referit3D [2] | | non-SAT [52] | |
|---|---|---|---|---|---|---|
| | | | Ref. Acc. | Cls. Acc | Ref. Acc. | Cls. Acc |
| - | - | 3D | 35.6 | 57.4 | 37.7 | 59.1 |
| 32 | 1 | | 28.0 | 47.34 | 30.9 | 49.31 |
| 64 | 1 | 2D | 32.0 | 56.93 | 33.7 | 56.14 |
| 128 | 1 | | **34.4** | **62.15** | **35.5** | **63.32** |
| 32 | 3 | | 29.7 | 50.11 | 31.1 | 51.40 |
| 64 | 3 | 2D | 32.9 | 63.20 | 34.2 | 64.02 |
| 128 | 3 | | **35.1** | **64.85** | **36.4** | **65.13** |
| 32 | 5 | | 31.1 | 55.18 | 33.7 | 56.72 |
| 64 | 5 | 2D | 34.1 | 64.05 | 35.6 | 64.52 |
| 128 | 5 | | **36.0** | **65.19** | **38.2** | **65.40** |

Table 3: Comparison between different types of 2D semantics information as in Eq. 1 on Nr3D [2].

| | $+x^{geo}$ | $+x^{cls}$ | $+x^{ROI}$ | SAT † [52] | | Ours | |
|---|---|---|---|---|---|---|---|
| | | | | Ref. Acc. | Cls. Acc | Ref. Acc. | Cls. Acc |
| (a) | - | - | - | 37.7 | 59.1 | **38.1** | 58.2 |
| (b) | - | - | ✓ | 42.3 | 61.0 | **42.2** | **62.2** |
| (c) | ✓ | - | ✓ | 46.2 | 61.0 | **46.3** | **63.3** |
| (d) | - | ✓ | ✓ | 43.2 | 61.2 | **44.5** | **62.1** |
| (e) | ✓ | ✓ | ✓ | 47.3 | 60.6 | **48.9** | **65.42** |

–**Network and Training Configuration.** A pre-trained model on the ImageNet dataset [9], Tiny-ConvNext [28], is used to extract the features of our synthetic 2D images. Our 2D backbone is shared across different views. We adopt the same 3D encoder in the previous works [1] [2] [13] [37] [18] [52], PointNet++ [35], to encode the sampled points per proposal. For the language encoder, we adopt the BERT encoder [11] with three layers. We adopted the same transformer architecture used in [52] [51] for our visual, multi-modal, and fusion transformers. All models are trained for 100 epochs from scratch, using the weight initialization strategy described in [14]. As mentioned in SAT [52], the initial learning rate is set to $10^{-4}$ and decreases by 0.65 every ten epochs . Adam optimizer [21] and mini-batch size of 32 per GPU are used for training all our models. We set the losses weights as follows, $\lambda_{cls} = 5$, $\lambda_{Eref} = 0.5$, $\lambda_{ref} = 5$, $\lambda_{cor} = 10$, and $\lambda_{lang} = 0.5$.

–**Software and Hardware Details.** We use the python augmentation framework, Albumentations [4], for image augmentation. In order to perform an online augmentation while training, we implemented our 2D Synthetic Images Generator (SIG), described in Section 3.1, in a pure Pythonic approach without using any 3D libraries, which also speedup the training. Our module is implemented in Python using the PyTorch framework and four Nvidia V100 GPUs.

## 4.4 Ablation studies

We conducted five ablation studies to assess the proposed architecture and to validate the contribution of each component to achieve good performance. These components are 1) 2D multi-views and 2D resolution. 2) Types of 2D semantic information. 3) Camera vs. image augmentation 4) SIG Module vs. 2D real clues. 5) Noisy Cameras Extrinsic and Intrinsic. More ablations are stated in the Appendix.

–**2D multi-views and 2D resolution.** To assess our 2D Synthetic Images Generator (SIG), we have deliberately removed the 3D stream, colored with yellow in Figure 3; 3D encoder, 3D heads, and rely only on our generated 2D images only while tackling the referring task. Thus, our architecture is simplified to include only the 2D encoder, 2D multi-modal transformer, and 2D heads. We trained this 2D architecture variant with different image resolutions and multi-view setup combinations. As shown in Table 2, The referring accuracy increased linearly while increasing the image size or increasing the number of views for the projected images. However, our multi-view setup has the dominant effect, whereas using only 64 image resolution with five views surpassing one shot with 128 resolution. Also, we found that using the combination of five different views, each has 128 resolution, surpassing Referit3D and non-SAT variant, where both use 3D semantics only, which shows the quality of the learned representation via our novel synthetic 2D images generator (SIG).

–**Types of 2D semantic information.** Inspired by SAT [52], we dissect the role of different 2D semantic, following Eq. 1. As shown in Table 3, the first row (a) represents a 3D variant of our architecture where only the 3D encoder, 3D multi-modal transformer, and 3D heads are included. Therefore, this variant mimics the Non-SAT [52] configurations. Starting from row b to row e, our synthetic 2D semantics are incorporated alongside the 3D semantics. Also, we compare against SAT [52] to highlight the critical differences in the meaning of each semantic information. ROI in our architecture is the synthetic 2D image generated from the 3d point clouds using our SIG module. In contrast, SAT [52] uses the original ScanNet images during the training phase. Unlike SAT, which uses the ground-truth classification labels during the training and masks them during testing, we use

Table 4: Comparison between Camera augmentation and Image augmentation on Nr3D [2].

| Cam. Aug. | Img. Aug. | Ref. Acc. | Cls. Acc. |
|:---:|:---:|:---:|:---:|
| ✗ | ✗ | 45.9 | 60.33 |
| ✗ | ✓ | 48.2 | 62.69 |
| ✓ | ✗ | 47.8 | 63.97 |
| ✓ | ✓ | **48.9** | **65.42** |

Table 5: Comparison between our SIG module and the real 2D clues utilized by SAT on Nr3D [2].

| Method. | Training | Inference | Ref. Acc. |
|:---:|:---:|:---:|:---:|
| SAT | Real 2D Images | ✗ | 47.3 |
| SAT | Real 2D Images | Real 2D Images | 47.9 |
| LAR | SIG | SIG | **48.9** |
| LAR | Real 2D Images | Real 2D Images | 46.8 |

Table 6: Showing LAR robustness against camera augmentation.

| Training | Testing | Ref. Acc. | Cls. Acc. |
|:---:|:---:|:---:|:---:|
| ✗ | ✗ | 48.2 | 62.69 |
| ✗ | ✓ | 46.7 | 61.32 |
| ✓ | ✗ | **48.9** | **65.42** |
| ✓ | ✓ | 48.8 | 65.22 |

Table 7: Benchmarking results on ScanRefer [5].

| Method | Ref. Acc. |
|:---:|:---:|
| Refit3D | 46.9 |
| non-SAT | 48.2 |
| SAT | 53.8 |
| SAT † | 52.3 |
| Ours | **54.6** |

Table 8: Benchmarking results on Nr3D [2], while using the ground truth classification.

| Method | Ref. Acc. | Cls. Acc. |
|:---:|:---:|:---:|
| SAT-GT | 50.3 | 61 |
| Ours | **62** | **64** |

the predicted labels from the 2D classification head $z'$. Accordingly, we do not need to mask the 2D semantics, as we are using synthetic ones. As shown in row (e), the best accuracy is achieved when we jointly incorporate the three semantic types. We also notice a significant superiority in classification accuracy where our synthetic 2D images assist the 3D classifier in both phases; training and testing.

–**Camera vs. Image Augmentation.** We have two kinds of augmentation during training: 1) Image augmentation. 2) Camera augmentation.

For image augmentation, we followed the conventional techniques used by the major computer vision applications like horizontal and vertical flipping, blurring, color jitter, random scaling, random shifting, and random cropping. For camera augmentation, we range $d^f$ from 1.5 to 4 meters, $d^{up}$ from 0.5 to 2.5 meters, and $\epsilon$ from 2% to 15%. By changing the variables mentioned above, the extrinsic camera parameters are augmented. In addition, random noise is added to the intrinsic camera parameters. Table 4 demonstrates the effect of each augmentation type alone and the effect of fusing both of them. The best accuracy is achieved by utilizing the two augmentation types.

–**SIG Module vs. 2D real clues.**

To validate our SIG module, we tailored a variant of our architecture, where our SIG module is replaced by the actual 2D clues produced by Faster R-CNN [39]. As shown in Table 5, replacing our SIG module with the features produced by Faster R-CNN [39] degrades the performance by 2%. In addition, even when we compared unfairly to methods that use additional 2D information, even the variant of SAT, which utilize real images during training and testing (row #2), we still outperform it by 1%.

–**Noisy Cameras Extrinsic and Intrinsic.** Due to the significant improvement introduced by utilizing the camera augmentation, as shown in Table 4, a detailed analysis is conducted to explore the robustness against the camera's variations. Consequently, our 2D synthetic branch learns how to fuse information across different cameras in a data-driven way. Therefore, we can train the model to be robust to noise camera models, as shown in Table 6.

Both cameras dying and noisy parameters experiments could be thought as irrelevant as LAR utilizes virtual cameras. Thus, no camera dropout nor noise can be introduced. However, our SIG module, depicted in Section 3.1, can be utilized in a wide range of applications when these robustness characteristics against such noise are essential for these applications. For instance, in the autonomous driving world, our SIG module can be employed to convert dense lidar point clouds into synthetic images, which will be used to train an arbitrary neural network that tackles specific tasks (e.g., object segmentation in cocoon setup [31] [47]). Then, the model will be tested using real cameras instead of generating synthetic imagery input from the lidar.

Table 9: Benchmarking results on Nr3D and Sr3D datasets [2]. All reported techniques are using the ground-truth object proposals. Additional input column emphasize the extra information used besides object's point cloud, where $S_{pc}$ and $2D_{Img}$ indicate the over all scene point clouds and extra 2D images, respectively. The † symbol indicates we retrain the model for a fair comparison.

| Method | Additional Input | Nr3D | | | | | Sr3D | | | | |
|---|---|---|---|---|---|---|---|---|---|---|---|
| | | Overall($\sigma$) | Easy | Hard | View-dep. | View-indep. | Overall($\sigma$) | Easy | Hard | View-dep. | View-indep. |
| ReferIt3D [2] | - | 35.6 | 43.6 | 27.9 | 32.5 | 37.1 | 40.8 | 44.7 | 31.5 | 39.2 | 40.8 |
| Text-Guided-GNNs [17] | - | 37.3 | 44.2 | 30.6 | 35.8 | 38.0 | 45.0 | 48.5 | 36.9 | 45.8 | 45.0 |
| InstanceRefer [58] | $S_{pc}$ | 38.8 | 46.0 | 31.8 | 34.5 | 41.9 | 48.0 | 51.1 | 40.5 | 45.4 | 48.1 |
| 3DRefTransformer [1] | - | 39.0 | 46.4 | 32.0 | 34.7 | 41.2 | 47.0 | 50.7 | 38.3 | 44.3 | 47.1 |
| 3DVG-Transformer [59] | - | 40.8 | 48.5 | 34.8 | 34.8 | 43.7 | 51.4 | 54.2 | 44.9 | 44.6 | 51.7 |
| FFL-3DOG [12] | - | 41.7 | 48.2 | 35.0 | 37.1 | 44.7 | - | - | - | - | - |
| TransRefer3D [13] | - | 42.1 | 48.5 | 36.0 | 36.5 | 44.9 | 57.4 | 60.5 | 50.2 | 49.9 | 57.7 |
| LanguageRefer [37] | - | 43.9 | 51.0 | 36.6 | 41.7 | 45.0 | 56.0 | 58.9 | 49.3 | 49.2 | 56.3 |
| Non-SAT [52] | - | 37.7 | 44.5 | 31.2 | 34.1 | 39.5 | 43.9 | - | - | - | - |
| SAT [52] | $2D_{Img}$ | 49.2 | 56.3 | 42.4 | 46.9 | 50.4 | 57.9 | 61.2 | 50.0 | 49.2 | 58.3 |
| SAT † [52] | $2D_{Img}$ | 47.3 | 55.8 | 41.4 | 46.1 | 49.9 | 56.6 | 60.6 | 49.7 | 48.7 | 57.4 |
| LAR (Ours) | - | **48.9**±0.2 | **56.1** | **41.8** | **46.7** | **50.2** | **59.35**±0.1 | **63.0** | **51.2** | **50.0** | **59.1** |

## 4.5 Comparison to State-of-the-art

Moreover, we achieve state-of-the-art results on the three well-known 3D visual grounding benchmarks, i.e., Nr3D [2], Sr3D [2], and ScanRefer [5]. By effectively utilizing our 2D synthetic images, LAR outperforms all the existing methods, which do not use any additional training dataset, by a large margin of 5.0% and 1.9% on Nr3D and Sr3D, respectively, depicted in Table 9. Where, LanguageRefer [37] achieves 43.9%, while LAR achieves 48.9% on Nr3D. For Sr3D, our model achieves 59.35%, which is a 1.9% increase over TransRefer3D [13]. In addition, for the ScanRefer [2] benchmark, we also achieve state-of-the-art results, by boosting the accuracy by 2.3% from the closest counterpart architecture; SAT [52], mentioned in Table 7.

We retrained SAT [52] for a fair comparison, as we notice difficulties reproducing and verifying SAT results [1]. Although we do not use an extra input during training as SAT, which uses the original ScanNet 2D images [8], we surpass SAT by 1.6%, 2.7%, and 2.3% on Nr3D, Sr3D, and ScanRefer, respectively.

## 4.6 Discussion and Analysis

First, we will demonstrate our qualitative analysis. Then, we conducted two experiments to probe the robustness of the representations learned by our proposed approach LAR.

–**Qualitative Analysis.** Figure 5 depicts two corner cases in our projection module (SIG). While assigning the projected 3D points into a 2D grid, more than one point may share the exact spatial location on the grid, which means only one will be stored in the grid, and the rest of the points will be discarded. As shown in part (A) in Figure 5, on the left column, the floor overlaps the other objects due to the conflict between points during the assignation step. To overcome this behavior, a simple priority rank is given for each point based on its height. Part (B) shows another failure case, where the prominent face is falsely localized, which causes placing the camera outside the room or inside other objects. Therefore, we localize the prominent face relative to the room center to mitigate this effect. An example of an ambiguous description is shown in Figure 6 part A. The model incorrectly predicts "the chair on the right rather than the chair on the left." The model and the human annotations are occasionally confusing for referring to the objects in the view-dependent description. While successful cases can be seen in Figure 6 part B. For more qualitative visualization, please refer to the Appendix.

–**Evaluation with Ground-Truth Class Labels.**

Table 8 shows another variant, which is tailored from our architecture to be compared against SAT-GT. Where the predicted class labels are replaced by the ground-truth ones. Our LAR architecture outperforms SAT-GT by 12%.

–**Zero-shot Cameras Dying Testing.** We deliberately drop the cameras during testing to verify the robustness of our learned representations

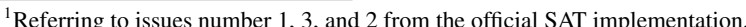

Figure 4: Cameras dropout.

[1]Referring to issues number 1, 3, and 2 from the official SAT implementation.

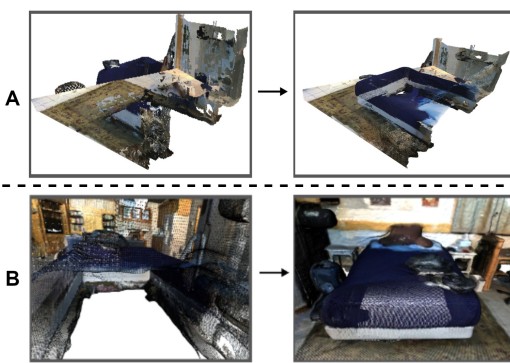

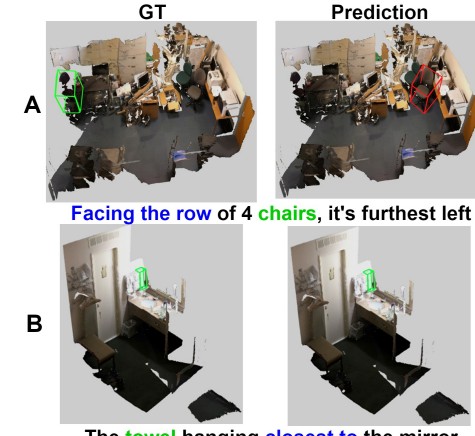

GT          Prediction

A

**Facing the row** of 4 **chairs**, it's furthest left

B

The **towel** hanging **closest to** the mirror

Figure 5: Qualitative results for projection corner cases. Part A shows failure in assignation step for the projected 3D points into the 2D grid. Part B, demonstrates failure in the prominent face localization step.

Figure 6: Our qualitative results. Case A shows a failure due to view-dependent description, While case B shows a successful case.

against camera variations. As shown in Figure 4, starting from our best model, which is trained using five cameras, the referring accuracy drops to 44.6% then 40.15%, when we only use three cameras then one camera during testing, respectively. Driven by these results, camera dropout techniques can be invoked to overcome the performance drop, by deliberately drop a random camera during the training; replace its input by a duplicate view of another camera.

–**Limitations.** Following the standard setup of Referit3D [2], we assume access to the object proposals of each 3D scene, similar to SAT [52], LanguageRefer [37], TransRefer3D [13], and 3DVG-Trans [59]. Therefore, we use the ground-truth proposals in our experiments. Following this assumption, the 3D grounding problem is reformulated as a classification problem, where the model objective is to classify the referred object among the other distractors correctly. However, LAR is compatible with the setting that uses detected-based proposals.

## 5 Conclusion

We propose an efficient 3D grounding architecture that leverages 2D knowledge to reinforce the learned 3D representations. To avoid requiring 2D semantics neither during training nor testing, we introduce a 2D synthetic image generator, termed SIG. Our SIG module generates high-quality 2D synthetic images. LAR, 2D Synthetic Semantics Knowledge Distillation for 3D Visual Grounding, consists of two stages, i.e., Visual and Multi-Modal modules. LAR preserves the advantages of distilling the synthetic 2D knowledge to the 3D stream without requiring any additional information. Our experiments show a significant superior performance, where LAR achieves state-of-the-art results across three different 3D grounding benchmarks by outperforming the existing approaches, which do not rely on extra training data, by a large margin, i.e., 5.0%, 1.9% and 2.3% on Nr3D, Sr3D and ScanRefer, respectively. Hence, despite the unfair comparison with SAT, which uses extra 2D images, LAR surpasses SAT by 1.6%, 2.7%, and 2.3% on Nr3D, Sr3D, and ScanRefer, respectively. Furthermore, we verify the robustness of LAR and its generalization ability via zero-shot experiments to intrinsic and extrinsic camera parameters and dying cameras. More qualitative results are attached in the supplementary materials.

## Acknowledgments and Disclosure of Funding

This work is funded by KAUST BAS/1/1685-01-0 and is also partially supported by the SDAIA-KAUST Center of Excellence in Data Science and Artificial Intelligence (SDAIA-KAUST AI).

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
