# Look Around and Refer: 2D Synthetic Semantics Knowledge Distillation for 3D Visual Grounding

**Eslam Mohamed Bakr, Yasmeen Alsaedy, Mohamed Elhoseiny**
King Abdullah University of Science and Technology (KAUST)
{eslam.abdelrahman, yasmeen.alsaedi, mohamed.elhoseiny}@kaust.edu.sa

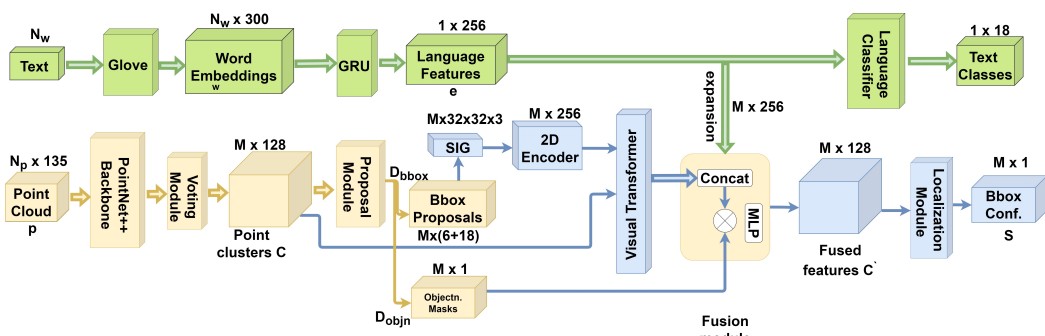

Figure 1: Detailed overview of the tailored variant, called LAR-Detection, which is adapted from ScanRefer. The proposals are generated using a 3D detector. The modules which are marked green or yellow indicate the pre-trained modules, while the blue ones are the integrated modules from our LAR architecture which is trained from scratch.

## A    Appendix

Our Supplemental Material contains the following sections:

- Detector-Generated 3D Proposal.
- Computational Cost.
- Demonstrate our synthetic images generator algorithm (SIG).
- Implementation Details.
- Evaluation Metrics
- More ablations.
- Qualitative results.

### A.1    Detector-Generated 3D Proposal

Following the standard setup of Referit3D [1], we assume access to the object proposals of each 3D scene, similar to SAT [5], LanguageRefer [4], TransRefer3D [3], and 3DVG-Trans [6]. Therefore, we use the ground-truth proposals in our experiments. Following this assumption, the 3D grounding problem is reformulated as a classification problem, where the model objective is to classify the referred object among the other distractors correctly. However, LAR is compatible with the setting that uses detected-based proposals. To this end, we experimented our LAR module with the detector-generated 3D proposals using VoteNet detector. As shown in Figure 1, we adapted the Scanrefer architecture as follows: 1) 2D Encoder: Integrating a 2D encoder right after the predicted box proposals, i.e., SIG module and Tiny-ConvNext in our case, and Faster R-CNN [1] in SAT case. 2) Fusion Transformer: In LAR case, we added our visual transformer only as we aim to emphasize the

Table 1: 3D visual grounding accuracy on ScanRef [2] with detector-generated proposals.

| Method | Acc@0.25 | Acc@0.5 |
|--------|----------|---------|
| ScanRefer | 40.92 | 26.08 |
| SAT | 39.2 | 26.25 |
| LAR | **42.14** | **26.96** |

contribution of our SIG module only. While in SAT case, we convert their multi-modal transformer to vision transformer by excluding the language embedding from it. 3) The same 3D prediction modules and language modules are adopted from Scanrefer, i.e., P++ and Voting module, and GloVe and GRU, respectively. We train three models: 1) The best variant of Scanrefer. 2) SAT-Detection-based architecture. 3) LAR-Detection-based architecture.

We first train the Scanrefer for 50 epochs then freeze their 3D prediction modules and language modules for two epochs to train the new blocks added in SAT and LAR detection based architectures. Then, we train the whole architecture for two more epochs. For more details, please refer to the "Detector-Generated 3D Proposal" section in Appendix A in the revised paper version, where we added a figure to demonstrate the adaptations done on the ScanRefer architecture.

As shown in Table 1, the tailored LAR-Detection-based architecture outperforms both the original ScanRefer and the SAT-Detection-based architectures. Also, the results show that SAT is more vulnerable to the distortions in the detected proposals.

## A.2 Computational Cost

We compared our complexity against SAT, during both the training and testing. As shown in Table 2, our training and inference time are 3.6 x and 0.7 x compared to SAT. All the reported results are measured on single GTX-1080 GPU.

## A.3 Ablations Studies

### A.3.1 Different 2D semantic fusion techniques.

Driven by the aforementioned results in the main paper, we use five virtual cameras set in a cocoon setup around each object, and the image resolution is 128. In Table 3, we study two fusion techniques to combine the features generated from the five views. The first method is a simple addition operation, and the second one is a learnable fusion mechanism where a 1D-Conv. is used where its kernel size is set to the number of the views used, i.e., in our case set to five. As expected, the learnable fusion scheme; 1D-Conv., performs better than the none learnable scheme; addition. Our 2D variant built on top of Referit3D [1] is used in this ablation, where the 2D stream is only used.

Table 3: Comparison between different 2D semantic fusion techniques.

| Method | Referring Acc. | Classification Acc. |
|--------|----------------|---------------------|
| Addition | 35.3 | 62.7 |
| 1D-Conv. | **36** | **64.82** |

Table 4: Comparison between different multi-modal fusion techniques.

| Method | Referring Acc. | Classification Acc. |
|--------|----------------|---------------------|
| Addition | 38.8 | 60.3 |
| 1D-Conv. | 39.2 | 60.8 |
| Transformer | **41.9** | **62.2** |

Table 2: Comparison between our LAR architecture against SAT in terms of number of parameters and time of both training and testing phases.

| Method | Ref. Acc. | Trainable Parameters | Inference Parameters | Training Time | Inference Time |
|--------|-----------|----------------------|----------------------|---------------|----------------|
| SAT | 47.3 | 237 M | **81 M** | 0.317 FPS | **5.97 FPS** |
| LAR | **48.9** | **118 M** | 118 M | **1.152 FPS** | 4.12 FPS |

### A.3.2 Different multi-modal fusion techniques.

In addition, we study three fusion techniques for the 3D, and 2D multi-modal features, $M^{pc}$ and $M^{img}$, respectively. In this ablation, we only utilize the ROI as the 2D semantic information, set the image resolution to 32, and set the number of multi-views to five. As shown in Table 4, using a transformer while fusing both multi-modal features achieves the best accuracy.

Optionally include extra information (complete proofs, additional experiments and plots) in the appendix. This section will often be part of the supplemental material.

## A.4 Synthetic Images Generator Algorithm

Algorithm 1, demonstrates a pseudo-code for our SIG module that projects arbitrary 3D Point clouds representing an object to a 2D image.

---

**Algorithm 1** 2D Synthetic Images Generator algorithm

---

**Input**: $x_i^{pc}$: Point-cloud for an object $i$ in the scene,
     $I$: The pinhole camera intrinsic parameters,
     $O_s$: Scene center.
**Output**: $x_i^{img}$: 2D synthetic image that represents the object $i$.

1:  // Get the prominent face:
2:  $minDist \leftarrow Dist(O_{l_0}, O_s)$
3:  **for** each face **do**
4:     **if** $Dist(O_{l_k}, O_s) <= minDist$ **then**
5:        $l' \leftarrow l_k$
6:        $minDist \leftarrow Dist(O_{l_k}, O_s)$
7:     **end if**
8:  **end for**
9:  $Cam_p = [d_x^f, d_y^f, d^{up}]$
10: // Build transformation matrix $M \in \mathbb{R}^{4 \times 4}$
11: $V^f \leftarrow \|Cam_p - O_{l'}\|$
12: $V^r \leftarrow \|V^f \otimes [0, 0, 1]\|$
13: $V^{up} \leftarrow \|V^r \otimes V^f\|$
14: $M \leftarrow \begin{bmatrix} V^r & V^{up} & V^f & Cam_p \\ 0 & 0 & 0 & 1 \end{bmatrix}$.
15: $x_i^{cam} \leftarrow Dot(M, x_i^{pc})$
16: $x_i^{img} \leftarrow Dot(I, x_i^{cam})$
17: $x_i^{img} \leftarrow x_i^{cam} / x_i^{cam}[:, 2]$
18: **return** $(x_i^{img})$

---

## A.5 Quantitative Results

This section analyzes the qualitative results from the successful and the failure cases on the Nr3d dataset. In addition to Figure 2 that shows successful scenarios and Figure 3 that shows failure scenarios, we attach a short video to demonstrate more scenarios.

### A.5.1 Successful Scenarios

LAR achieves top-ranking results on both the Nr3D and Sr3D datasets. The results in Figure 2 demonstrate the correct instance localization on the Nr3D datasets only. The ground truth boxes are marked green and our model prediction marked in blue. Our analysis shows that LAR correctly understands the global context based on natural language description. For example, in Figure 2 (a-f), LAR understands relationships, e.g. (behind it, middle of, furthest from .. ), which proves that our proposed model understands instance-to-instance and instance-to-background relations. As well as understanding colors and shapes related to language, LAR also can determine the meaning of terms such as "green," Figure 2 (e).

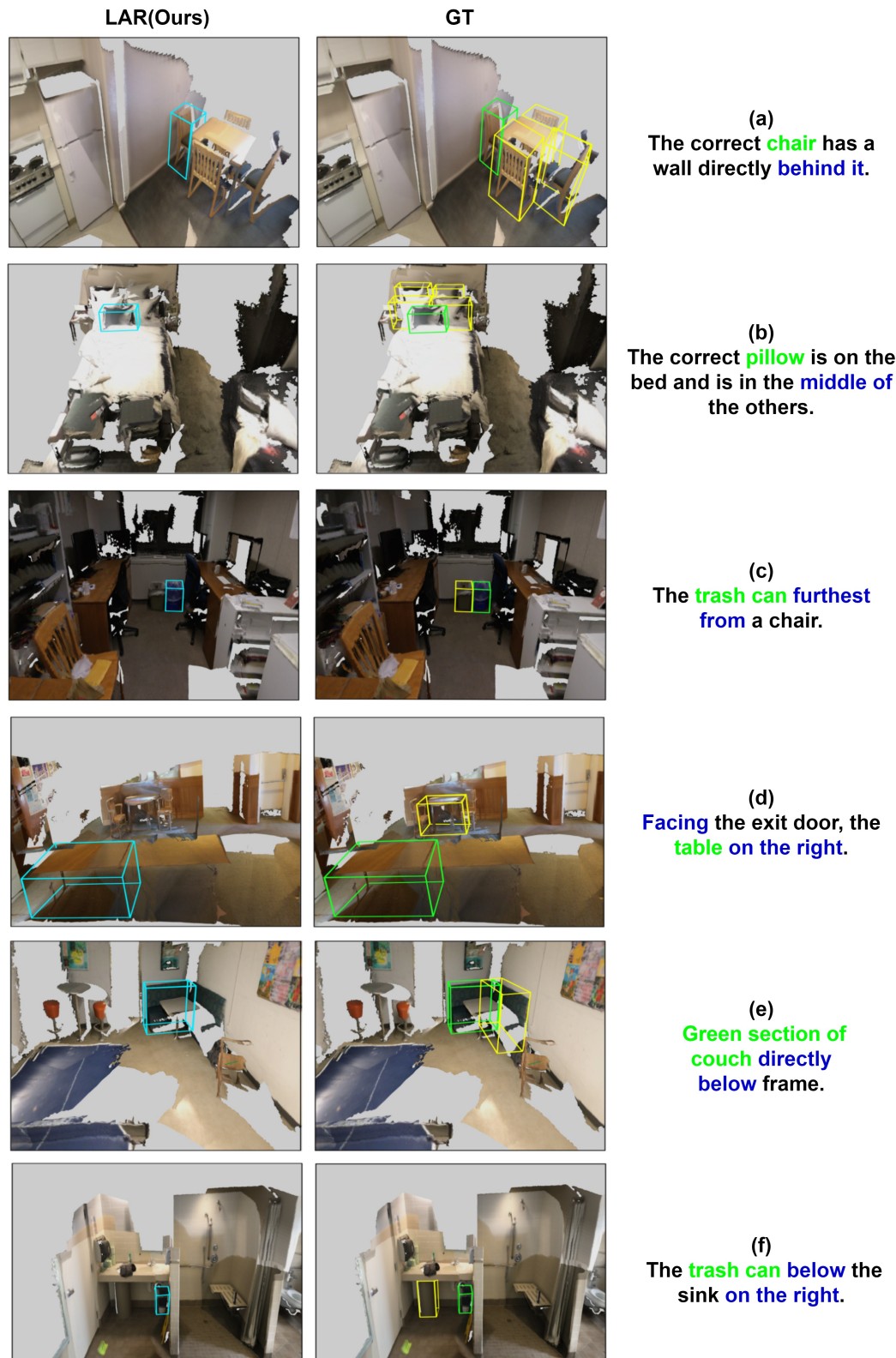

Figure 2: Qualitative results of successful cases from our LAR on Nr3d. The ground truth boxes are marked green, and the distractors boxes are displayed in yellow. Our model prediction marked as blue.

### A.5.2  Failure Scenarios

Figure 3 shows representative failure cases of LAR. Despise the incorrect instance localization LAR managed to allocate the instance with the same class as the target. As shown in Figure 3 (d, e), we observed that failure cases require an understanding of "against the wall", 'Facing the windows, "which is determined by camera view dependency. Despite LAR's progress in view-dependent and independent samples, view understanding remains a challenge. Besides, other state-of-the-art models suffer from the same issue. In addition, some LAR failure cases are caused by ambiguous queries. Figure 3 (a,b,c,e, f) shows a complex and compound language description to describe the target instance, which in some cases, both models and human annotators mix up similar objects.

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

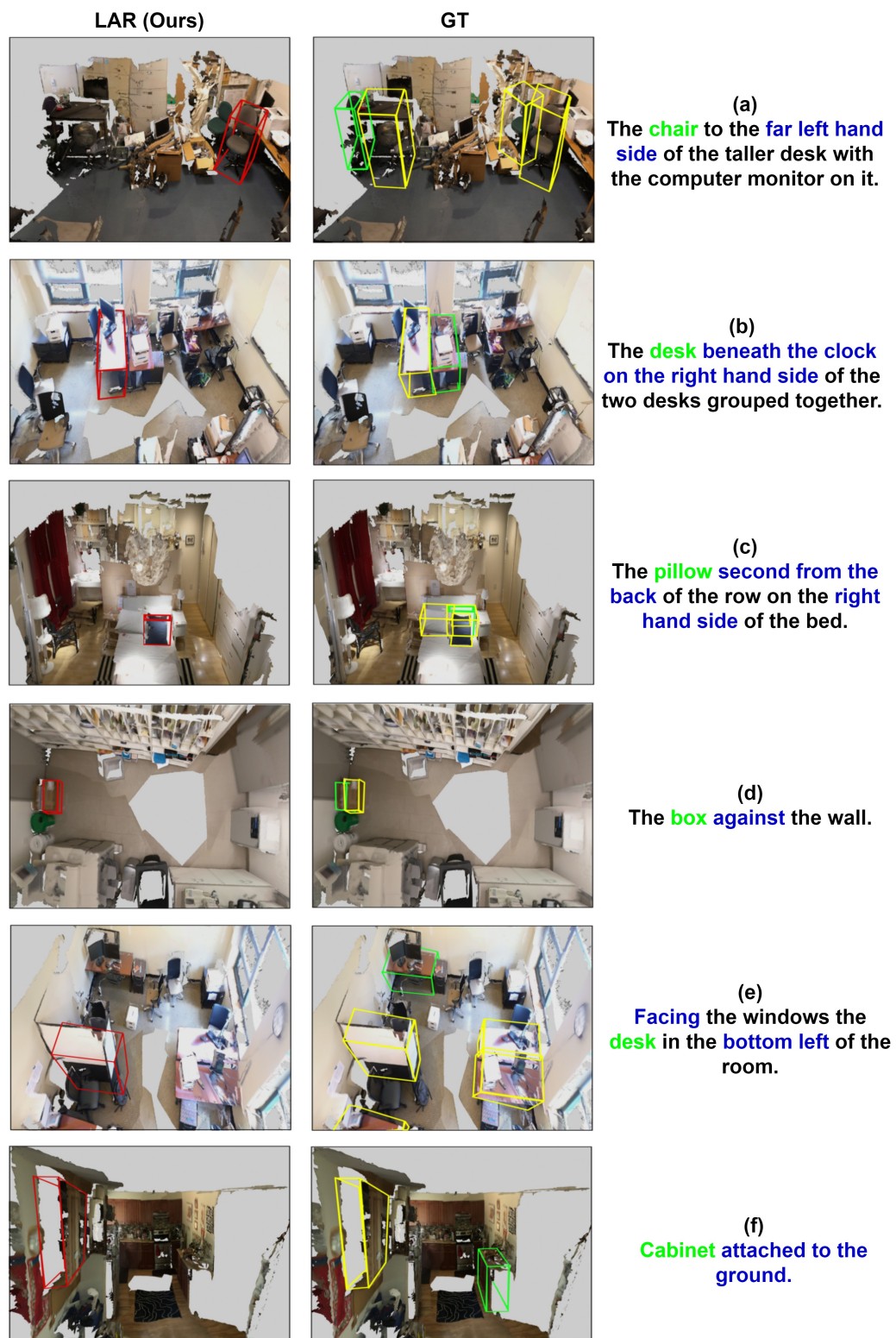

LAR (Ours)       GT

**(a)**
The **chair** to the **far left hand side** of the taller desk with the computer monitor on it.

**(b)**
The **desk beneath the clock on the right hand side** of the two desks grouped together.

**(c)**
The **pillow second from the back** of the row on the **right hand side** of the bed.

**(d)**
The **box against** the wall.

**(e)**
**Facing** the windows the **desk** in the **bottom left** of the room.

**(f)**
**Cabinet attached to the ground.**

Figure 3: Qualitative results of failure cases from our LAR on Nr3d. The Ground truth boxes are shown in green, and the distractors boxes are displayed in yellow. The incorrect predictions by our model marked as red.