# OpenReview forum: "Look Around and Refer: 2D Synthetic Semantics Knowledge Distillation for 3D Visual Grounding"
_NeurIPS.cc/2022/Conference — NeurIPS 2022 Accept_

### Official Review · Reviewer_8Se9 · 2022-06-17

**Rating:** 5
**Confidence:** 4
**Soundness:** 3 good
**Presentation:** 2 fair
**Contribution:** 3 good

**Summary:**

This paper studies 3D Visual Grounding. Its goal is to identify from a 3D point cloud the 3D object referred in a sentence. The idea in this paper is to utilize RGB image as another input modality to help the grounding task. Different from the previous method SAT [51], the RGB info in the proposed method is obtained by rendering the RGB point cloud from multiple viewpoints, while SAT [51] obtains it from camera input from one single view. The paper further proposes a novel architecture to fuse the rendered RGB input and the point cloud, leading to better performance than SAT [51].

**Questions:**

Overall I feel that a lot of details are missing from the paper. I think adding those details are necessary to make the paper self-contained and easier to be understood. For example:
1. Where do the 3D object proposals come from during training and inference?  How do you know what is the upper and lower faces of the bounding box as stated in line 136?
2. Where does the predicted label z'_i in line 197 come from? From a trained image classifier? How is it trained? How many classes is it producing?
3. What is the "language classification head" in line 203? What is its purpose?
4. How exactly are the losses in line 213 to 220 defined?


I am also curious to know what the performance would be if the proposed method uses the same RGB input as SAT, i.e. one single image from camera. That would make the proposed SIG more justified.

**Limitations:**

The authors have adequately addressed the limitations and potential negative societal impact of their work

**Strengths And Weaknesses:**

Strengths:
+ The paper designs a clever rendering pipeline (SIG) to render the input point cloud into multiple images, which are fed into the network together with the point cloud. This strategy provides more RGB signal than pure camera inputs, and together with the novel network that fuses multi-modal inputs, the proposed method outperform SAT (which requires an RGB camera input). This is an effective strategy, and as the number of provided view increases, the performance increases (Table2).


Weaknesses:
- I think the strategy designed to render the point cloud seems debatable, and it is possible that there is another object blocking the path from the camera center to the proposal object, resulting in occlusion. In this case the rendered image would be incorrect.
- It seems that the proposed method requires a trained image classifier? If that is the case wouldn't that make the comparison with SAT [51] unfair because more information is given to train this framework?

---

> ### Author Response · Authors · 2022-08-03
> **Author response to Reviewer 8Se9 [Part 1]**
>
> We thank you for your valuable and thoughtful feedback.
> We are encouraged that you find our approach novel, effective, well motivated and clever.
> In addition to finding our experiments comprehensive and outperforming SOTA.
> Also, we highly appreciate your improvement suggestions.
> Below, we will address your concerns and questions and incorporate all the feedback.
>
> # ---------------------------------------------------------
> **1: I think the strategy designed to render the point cloud seems debatable, and it is possible that there is another object blocking the path from the camera center to the proposal object, resulting in occlusion. In this case the rendered image would be incorrect.**
>
> Thanks for pointing out this corner case in designing our SIG module.
> Our SIG module is robust enough against such occlusion scenarios, thanks to accumulating the clues from more than one view.
> Our multi-view setup enables us to overcome the occlusion caused by one view by leveraging the other views while constructing our 2D grid.
>
> # ---------------------------------------------------------
> **2: It seems that the proposed method requires a trained image classifier? If that is the case wouldn't that make the comparison with SAT [51] unfair because more information is given to train this framework?**
>
> Yes, our proposed architecture, termed LAR, uses a pre-trained image classifier, i.e., Tiny-ConvNext pre-trained on the ImageNet dataset [5].
> However, that does not indicate LAR leverages extra information than SAT, as SAT requires an instant search retrieval system, such as Faster R-CNN [4], to sample the most suitable 2D clues/learned representation from the object bounding boxes.
>
> Also, The instant search retrieval system [4] utilized by SAT is pre-trained on the Visual Genome dataset [7], which is much bigger than the ImageNet dataset [5].
>
> Therefore, It could be interpreted as unfair to us as we don't use additional information, i.e., real 2D images, and we use a simpler pre-trained 2D encoder.
>
> # ---------------------------------------------------------
> **3: Where do the 3D object proposals come from during training and inference?**
>
> We followed the convention of the previous works, that assume the proposals are generated with GT objects.
>
> Please refer to pieces of evidence that the whole methods used in our benchmark are using GT-proposals for fair comparison:
>
> - SAT [6] authors stated it in Sec. 5.2.
>
> - InstanceRefer mentioned this in issue 4 [2].
>
> - 3DVG-Transformer authors mention it in issue 5 [3].
>
> - The rest of the architectures mentioned that explicitly in their publications.
>
> Following this assumption, the 3D grounding problem is reformulated as a classification problem, where the model objective is to classify the referred object among the other distractors correctly. However, LAR is compatible with the setting that uses detected-based proposals.
>
>
> # ---------------------------------------------------------
> **4: How do you know what is the upper and lower faces of the bounding box as stated in line 136?**
>
> Given an arbitrary 3D bounding box $B \in \mathbb{R}^{8\times 3}$, and the sensor location or the relative origin defined by the dataset itself, we sort the points concerning the Z dimension then split them in half. The four points that share the highest height will correspond to the upper face, while other points correspond to the lower face.
>
>
>
> **References:**
>
> [1] Referit3d: Neural listeners for fine-grained 3d object identification in real-world scenes.
>
> [2] https://github.com/CurryYuan/InstanceRefer/issues/4.
>
> [3] https://github.com/zlccccc/3DVG-Transformer/issues/5
>
> [4] Salvador, Amaia, et al. "Faster r-cnn features for instance search." Proceedings of the IEEE conference on computer vision and pattern recognition workshops. 2016.
>
> [5] Deng, Jia, et al. "Imagenet: A large-scale hierarchical image database." 2009 IEEE conference on computer vision and pattern recognition. Ieee, 2009.
>
> [6] Yang, Zhengyuan, et al. "Sat: 2d semantics assisted training for 3d visual grounding." Proceedings of the IEEE/CVF International Conference on Computer Vision. 2021.
>
> [7] Krishna, Ranjay, et al. "Visual genome: Connecting language and vision using crowdsourced dense image annotations." International journal of computer vision 123.1 (2017): 32-73.

---

> ### Author Response · Authors · 2022-08-03
> **Author response to Reviewer 8Se9 [Part 2]**
>
> # ---------------------------------------------------------
> **5: Where does the predicted label $z'_i$ in line 197 come from? From a trained image classifier? How is it trained? How many classes is it producing?**
>
> Yes, the 2D predicted label $z'_i$ is produced by the 2D classification head (colored in yellow in Figure 3).
> Consequently, agnostic proposals are fed to our 2D stream, where the 2D classification head aims to predict the object's class.
> Then, the 2D predicted labels $z'$ should be fused with the 2D image features $v^{img}$ produced by our visual transformer.
> But, Figure 3 shows by mistake that it is fused with $E^{2d}$.
> Thanks for pointing out this mistake.
> Please refer to the corrected Figure in the updated version.
>
> # ---------------------------------------------------------
> **6: What is the "language classification head" in line 203? What is its purpose?**
>
> The language classification accuracy assesses the performance of the linguistic branch. This branch aims to correctly predict the class of the referred object based on the input utterance.
>
> Alongside the main objective, which is the referring accuracy, two auxiliary metrics are defined to evaluate the performance of each stream individually, i.e., the visual classification accuracy and the language classification accuracy.
>
> # ---------------------------------------------------------
> **7: How exactly are the losses defined?**
>
> Our architecture, termed LAR, consists of two visual streams, i.e., 3D and 2D, in addition to one language stream.
> To refer to the object correctly guided by the input utterance, we must first classify the objects accurately.
> Thus, we utilize two auxiliary classification losses, i.e., 3D classification loss $\mathcal{L}_{cls}^{3D}$
>
> and 2D classification loss $\mathcal{L}_{cls}^{2D}$, to assist the model in learning the objects classes to improve the visual grounding task.
>
> We adopted Referit3D [1] auxiliary loss for language classification loss $\mathcal{L}_{cls}^{lang}$ to cooperate in the visual grounding task by recognizing the referred object class according to the description.
>
> Hence, two auxiliary grounding losses $\mathcal{L}_{Eref}^{3D}$
>
> and $\mathcal{L}_{Eref}^{2D}$, termed as early referring losses, are exploited before the fusion transformer.
> As a result of these early referring losses, the model learns the correlation between language description and the visual features separately, which enhances the performance.
>
> Additionally, we adopted object correspondence loss $\mathcal{L}_{cor}$
>
>  from SAT to encourage the 3D model to distill the knowledge from the 2D stream.
> Finally, two grounding losses, i.e., $\mathcal{L}_{ref}^{3D}$
>
>  and $\mathcal{L}_{ref}^{2D}$ were added after the fusion transformer to predict the scores of each proposal.
> As mentioned above, we assume access to the object proposals of each 3D scene; accordingly, the 3D grounding problem is reformulated as a classification problem.
>
> # ---------------------------------------------------------
> **8: I am also curious to know what the performance would be if the proposed method uses the same RGB input as SAT, i.e. one single image from camera. That would make the proposed SIG more justified.**
>
> To validate this, we tailored a variant of our architecture, where our SIG module is replaced by the actual 2D clues produced by Faster R-CNN [4].
> As shown in the table below, replacing our SIG module with the features produced by Faster R-CNN [4] degrades the performance by 2\%.
> In addition, even when we compared unfairly to methods that use additional 2D information, even the variant of SAT, which utilize real images during training and testing (row #2), we still outperform it by 1\%.
>
> | Method | Training       | Inference      |     Ref. Acc. |
> |--------|----------------|----------------|---------------|
> | SAT    | Real 2D Images |            X   |         47.3  |
> | SAT    | Real 2D Images | Real 2D Images |         47.9  |
> | LAR    |         SIG    |          SIG   |         48.9  |
> | LAR    | Real 2D Images | Real 2D Images |         46.8  |
>
> **References:**
>
> [1] Referit3d: Neural listeners for fine-grained 3d object identification in real-world scenes.
>
> [2] https://github.com/CurryYuan/InstanceRefer/issues/4.
>
> [3] https://github.com/zlccccc/3DVG-Transformer/issues/5
>
> [4] Salvador, Amaia, et al. "Faster r-cnn features for instance search." Proceedings of the IEEE conference on computer vision and pattern recognition workshops. 2016.
>
> [5] Deng, Jia, et al. "Imagenet: A large-scale hierarchical image database." 2009 IEEE conference on computer vision and pattern recognition. Ieee, 2009.
>
> [6] Yang, Zhengyuan, et al. "Sat: 2d semantics assisted training for 3d visual grounding." Proceedings of the IEEE/CVF International Conference on Computer Vision. 2021.
>
> [7] Krishna, Ranjay, et al. "Visual genome: Connecting language and vision using crowdsourced dense image annotations." International journal of computer vision 123.1 (2017): 32-73.

---

> ### Author Response · Authors · 2022-08-08
> **Author response to Reviewer 8Se9 [Part 3]**
>
> # ---------------------------------------------------------
> **[Updated-Experiments] 3: Where do the 3D object proposals come from during training and inference?**
>
> We followed the convention of the previous works, that assume the proposals are generated with GT objects.
>
> Please refer to pieces of evidence that the whole methods used in our benchmark are using GT-proposals for fair comparison:
>
> - SAT [6] authors stated it in Sec. 5.2.
>
> - InstanceRefer mentioned this in issue 4 [2].
>
> - 3DVG-Transformer authors mention it in issue 5 [3].
>
> - The rest of the architectures mentioned that explicitly in their publications.
>
> Following this assumption, the 3D grounding problem is reformulated as a classification problem, where the model objective is to classify the referred object among the other distractors correctly.
>
> However, LAR is compatible with the setting that uses detected-based proposals.
> We experimented our LAR module with the detector-generated 3D proposals using VoteNet detector.
> To this end, we adapted the Scanrefer architecture as follows:
> 1) 2D Encoder: Integrating a 2D encoder right after the predicted box proposals, i.e., SIG module and Tiny-ConvNext in our case, and Faster R-CNN [1] in SAT case.
> 2) Fusion Transformer: In LAR case, we added our visual transformer only as we aim to emphasize the contribution of our SIG module only. While in SAT case, we convert their multi-modal transformer to vision transformer by excluding the language embedding from it.
> 3) The same 3D prediction modules and language modules are adopted from Scanrefer, i.e., P++ and Voting module, and GloVe and GRU, respectively.
>
> We train three models:
> 1) The best variant of Scanrefer.
> 2) SAT-Detection-based architecture.
> 3) LAR-Detection-based architecture.
>
> We first train the Scanrefer for 50 epochs then freeze their 3D prediction modules and language modules for two epochs to train the new blocks added in SAT and LAR detection based architectures. Then, we train the whole architecture for two more epochs. For more details, please refer to the "Detector-Generated 3D Proposal" section in Appendix A in the revised paper version, where we added a figure to demonstrate the adaptations done on the ScanRefer architecture.
>
> As shown in the below table, the tailored LAR-Detection-based architecture outperforms both the original ScanRefer and the SAT-Detection-based architectures.
> Also, the results show that SAT is more vulnerable to the distortions in the detected proposals.
>
> |   Method  |    Acc@0.25    |     Acc@0.5    |
> |-----------|----------------|----------------|
> | ScanRefer |     40.41      |      26.08     |
> |    SAT    |     40.92      |      26.25     |
> |    LAR    |     **42.14**      |      **26.96**     |

---

### Official Review · Reviewer_yWd9 · 2022-07-09

**Rating:** 7
**Confidence:** 4
**Soundness:** 3 good
**Presentation:** 3 good
**Contribution:** 3 good

**Summary:**

The paper proposes a method to enhance the visual part of a 3D visual grounding system by integrating 2D semantic information. The system receives a 3D point cloud, a text query, and point subsets for the candidate objects as input. The paper proposes to render each candidate point set into multiple virtual cameras and use a transformer to fuse 2D and 3D information. The visual information is merged further with the text to classify the objects for finding the desired candidate.


**Questions:**

I have a couple of questions regarding the paper:
1. In the limitation section, the paper mentions using a detector instead of getting the ground truth point set proposals. How would this work with noisy detections? If points are cut out or added, this will drastically alter the information gained from the 2D encoder especially compared to a method like SAT, which uses ground truth 2D images. Also, if the goal is to use a robot, it would most likely be equipped with a camera and get a stream of 2D images. What was the rationale for not using this information and only using the synthetic 2D images instead? Line 110 mentions: "SAT [51 ] leveraged 2D semantics to perform the 3D VG task. However, requiring extra 2D inputs limits potential application scenarios." What limitations does the paper refer to here? As the motivation seems to be mainly for VR and robotics applications, where cameras are used either on the VR device or on the robot itself, are images not widely available?

2. The paper experiments show that it outperforms methods that have access to ground truth 2D images. How can this be explained? Are the single view images in the dataset taken from arbitrary angles? Is this paper just using a more advantageous camera angle? Or is this mainly due to the multi-view setting?

3. The paper focuses on indoor scenes by how it selects the viewing directions of the virtual cameras. How would this work in more open-world settings where the camera does not always point outwards? Would this approach still work?

**Limitations:**

The paper contains a concise section on the technical limitations, namely using ground truth candidate point sets (see my mention regarding this under questions). Negative societal impacts are not discussed at all.


**Strengths And Weaknesses:**

Strengths:
1. In my opinion, the overall technical idea of the paper, to use 2D information together with 3D information for visual grounding, is novel and exciting. In contrast to 3D point cloud encoders, 2D feature encoders can be pretrained on large datasets and aggregate more information. Generating synthetic images through point cloud rendering also generates multiple views per object, increasing the chance of generating a dominant view of the object.
2. The method achieves strong results in terms of classification accuracy, outperforming methods that use ground truth 2D images through rendering synthetic multi-view images.
3. I appreciated the qualitative examples in the supplementary material of both success and failure cases. Maybe showing qualitative results of SAT and adding a figure to the main paper would increase the readers understanding of the significance of the results.
4. The paper contains code and a supplementary video.

Weaknesses:
1. Section 3.1, lines 145 - 168. In my opinion, this section is overly wordy, explaining a well-known concept (pinhole projection), which takes space away from other parts of the paper and could use additional context. The paper explains how to project 3D points to a pinhole camera plane. As this is a very well-known topic in the literature, it suffices to mention which conventions are used for the extrinsic and intrinsic camera matrices and how the camera axes are defined. This should at most take a short paragraph and can even be shortened by using equations cleverly.

3. In contrast, section 3.2 does not contain enough information, in my opinion. Equation 1 contains the definition of how image information is fused with bounding box information, but this is merely stated as fact, without giving intuition about why this might be beneficial. Similarly, in lines 213-223, the paper presents the loss terms without a proper introduction. Most losses are not defined or only are merely referring to previous papers. In my opinion, this makes the paper hard to understand and not self-contained. Shortening section 3.1, the paper should contain enough space to add the missing details.

4. Section 3.3 Software Details: Here, the paper mentions for the first time that image augmentations were used, without specifying what kind of augmentations and why this is needed. The type of augmentations is mentioned later in Line 299, but there might be a better place.

5. Section 4: The paper does not introduce the used evaluation metrics. These should be contained within the paper so that the reader can evaluate what the quantitative results represent.

6. The paper does not contain a qualitative comparison of the detection performance with state-of-the-art. I am guessing that the quantitative score measures classification accuracy, but it is not apparent how large of a difference the improvement the method provides quantitatively.

Minor issues, spelling, and grammar:
* Figures 4, 5, and 6 are of different sizes and, therefore, are not aligned. Please align them such that the captions align.
* Table 6, Overall and hard performance on Nr3D. The paper marks its approach as best, although SAT performs slightly better. Please change this. I do not think it is an issue that the presented approach does not outperform the reported numbers from SAT, especially regarding the reproducibility issues. One of the paper's strong points is that the SAT approach has been retrained in this case.

I think the authors should address these weaknesses to polish the paper and make it more self-contained. If these issues are addressed, I am willing to increase my score.

---

> ### Author Response · Authors · 2022-08-03
> **Author response to Reviewer yWd9 [Part 1]**
>
> We thank you for your valuable and thoughtful feedback.
> We are encouraged that you find our approach novel, effective, well motivated, and our experiments comprehensive and outperforming SOTA.
> Also, we highly appreciate you thoroughly read the main manuscript and the supplementary materials.
> Below, we  address the concerns and will incorporate all the feedback.
>
> # ---------------------------------------------------------
> **1: This section is overly wordy, explaining a well-known concept (pinhole projection), which takes space away from other parts of the paper and could use additional context.
> The paper explains how to project 3D points to a pinhole camera plane. As this is a very well-known topic in the literature, it suffices to mention which conventions are used for the extrinsic and intrinsic camera matrices and how the camera axes are defined.**
>
> Thank you for the suggestions and for pointing out problems with the paper writing.
>
> We shortened Section 3.1 in the revised version and made it more apparent by reformulating its significant parts, marked in blue.
>
> # ---------------------------------------------------------
> **2: section 3.2 does not contain enough information, in my opinion. Equation 1 contains the definition of how image information is fused with bounding box information, but this is merely stated as fact, without giving intuition about why this might be beneficial.**
>
> Thank you for the suggestions and for pointing out problems with the paper writing.
> We added some clarifications in Section 3.2, marked in blue, to give adequate motivation for why we formulate our 2D semantics in such a way.
>
> In addition, here is the answer to your question: "why this might be beneficial?"
> We encode each object individually using different semantic information types, i.e., visual semantics (2D & 3D), and $x_i^{geo} \in \mathbb{R}^{1\times 30}$.
> The geometry information represents the spatial location of each object $i$ in the scene and encodes the virtual camera parameters.
> Thus, it can be interpreted as a positional encoding for the vision transformer, where the interaction among objects is captured.
>
> # ---------------------------------------------------------
> **3: lines 213-223, the paper presents the loss terms without a proper introduction. Most losses are not defined or only are merely referring to previous papers. In my opinion, this makes the paper hard to understand and not self-contained.**
>
> Thanks for your suggestion. We detailed the losses in Section 3.2 in the revised version to make the paper self-contained and clearer.
>
> Hence, here is the added section:
>
> Our architecture, LAR, consists of two visual streams, i.e., 3D and 2D, in addition to one language stream.
> To refer to the object correctly guided by the input utterance, we must first classify the objects accurately.
> Thus, we utilize two auxiliary classification losses, i.e., 3D classification loss
> $\mathcal{L}_{cls}^{3D}$
>
>  and 2D classification loss $\mathcal{L}_{cls}^{2D}$, to assist the model in learning the objects classes to improve the visual grounding task.
>
> We adopted Referit3D [5] auxiliary loss for language classification loss $\mathcal{L}_{cls}^{lang}$ to cooperate in the visual grounding task by recognizing the referred object class according to the description.
>
> Hence, two auxiliary grounding losses $\mathcal{L}_{Eref}^{3D}$
>
> and $\mathcal{L}_{Eref}^{2D}$, termed as early referring losses, are exploited before the fusion transformer.
> As a result of these early referring losses, the model learns the correlation between language description and the visual features separately, which enhances the performance.
>
> Additionally, we adopted object correspondence loss $\mathcal{L}_{cor}$
>
> from SAT to encourage the 3D model to distill the knowledge from the 2D stream.
> Finally, two grounding losses, i.e., $\mathcal{L}_{ref}^{3D}$
>
> and $\mathcal{L}_{ref}^{2D}$ were added after the fusion transformer to predict the scores of each proposal.
> As mentioned above, we assume access to the object proposals of each 3D scene; accordingly, the 3D grounding problem is reformulated as a classification problem.

---

> ### Author Response · Authors · 2022-08-03
> **Author response to Reviewer yWd9 [Part 2]**
>
> # ---------------------------------------------------------
> **4: Section 4: The paper does not introduce the used evaluation metrics.**
>
> Thanks for your suggestion. We mentioned the evaluation metrics in Section 4 in the revised version, colored in blue.
>
> Hence, here is the added section:
>
> Regarding the evaluation metrics, we followed the same metrics followed by the previous work (ReferIt3D, LanguageRefer, SAT, etc..), where three evaluation metrics are mainly used, i.e., the referring accuracy, the visual classification accuracy, and the language classification accuracy.
> In the case of Nr3d and Sr3d datasets, we followed the previous work convention, assuming the proposals are generated with GT objects.
> Following this assumption, the 3D grounding problem is reformulated as a classification problem, where the model objective is to classify the referred object among the other distractors correctly, termed referring accuracy.
> The referring accuracy is calculated based on whether the model picks the correct proposal from a set of X proposals.
> Alongside the main objective, which is the referring accuracy, two auxiliary metrics are defined to evaluate the performance of each stream individually, i.e., the visual classification accuracy and the language classification accuracy.
> Given agnostic GT-proposals, the visual classification accuracy is measured based on whether the model predicts the object's class correctly or not from a predefined set of possible classes (e.g., 524 class in the Nr3D dataset).
> Similarly, language classification accuracy is another auxiliary metric that assesses the performance of the linguistic branch. This branch aims to correctly predict the class of the referred object based on the input utterance.
>
> # ---------------------------------------------------------
> **5: The paper does not contain a qualitative comparison of the detection performance with state-of-the-art.**
>
> This is justified as the referring accuracy was calculated based on whether the model picks the correct proposal from a set of X proposals.
> As, we followed the convention of the previous works, that assume the proposals are generated with GT objects.
>
> Please refer to pieces of evidence that the whole methods used in our benchmark are using GT-proposals for fair comparison:
>
> - SAT [1] authors stated it in Sec. 5.2.
>
> - InstanceRefer mentioned this in issue 4 [2].
>
> - 3DVG-Transformer authors mention it in issue 5 [3].
>
> - The rest of the architectures mentioned that explicitly in their publications.
>
> Therefore, we didn't report qualitative results for the detection performance.
> However, we have conducted a detection-based experiment based on the ScanRefer dataset, where we don't assume access to the GT proposals, and once it is done, we will share the results with you.
>
> # ---------------------------------------------------------
> **6: I am guessing that the quantitative score measures classification accuracy, but it is not apparent how large of a difference the improvement the method provides quantitatively**
>
> First, yes, the quantitative score measures classification accuracy.
> As detailed above, the primary quantitative score measures the referring accuracy, which is interpreted as a classification accuracy due to the assumption mentioned above.
>
> Secondly, LAR exceeds all existing approaches that did not use additional information than the 3D point clouds by a significant margin, i.e., 5.0\% to the closest one.
> In addition, even when we unfairly compared methods that use additional 2D information, such as SAT, we still outperformed it by 1.6\%.
>
>
> **References:**
>
> [1] Yang, Zhengyuan, et al. "Sat: 2d semantics assisted training for 3d visual grounding." Proceedings of the IEEE/CVF International Conference on Computer Vision. 2021.
>
> [2] https://github.com/CurryYuan/InstanceRefer/issues/4
>
> [3] https://github.com/zlccccc/3DVG-Transformer/issues/5
>
> [4] Salvador, Amaia, et al. "Faster r-cnn features for instance search." Proceedings of the IEEE conference on computer vision and pattern recognition workshops. 2016.
>
> [5] Referit3d: Neural listeners for fine-grained 3d object identification in real-world scenes.

---

> ### Author Response · Authors · 2022-08-03
> **Author response to Reviewer yWd9 [Part 3]**
>
> # ---------------------------------------------------------
> **7: In the limitation section, the paper mentions using a detector instead of getting the ground truth point set proposals. How would this work with noisy detections? If points are cut out or added, this will drastically alter the information gained from the 2D encoder especially compared to a method like SAT, which uses ground truth 2D images.**
>
> We agree that the predicted proposals' quality considerably affects the referring accuracy; however, we argue that our approach is more robust than SAT against such distortions.
> Thanks to accumulating the clues from more than one view.
> Our multi-view setup enables us to overcome the occlusion or the distortion caused by one view by leveraging the other views while constructing our 2D grid.
>
> In comparison to SAT, it will hugely suffer from such cases, as SAT requires an instant search retrieval system, such as Faster R-CNN [4], to sample the most suitable 2D clues/learned representation from the object bounding boxes, which means that SAT performance will highly depend on the quality of the proposals.
>
> # ---------------------------------------------------------
> **8: Also, if the goal is to use a robot, it would most likely be equipped with a camera and get a stream of 2D images. What was the rationale for not using this information and only using the synthetic 2D images instead? Line 110 mentions: "SAT [51 ] leveraged 2D semantics to perform the 3D VG task. However, requiring extra 2D inputs limits potential application scenarios." What limitations does the paper refer to here? As the motivation seems to be mainly for VR and robotics applications, where cameras are used either on the VR device or on the robot itself, are images not widely available?.**
>
> First, we want to abide by the same setup followed by the preceding work for a fair comparison, where only 3D inputs are utilized while tackling the 3D visual grounding task.
> In addition, we aim to distill the 2D information in a more 3D-aware way by leveraging synthetic 2D information captured from different views, which is not leveraged in previous methods.
> To validate the power of our solution, we show that using real 2D clues, used by SAT, instead of our SIG module will degrade the performance by 2\%, as shown in the below table.
> In addition, even when we compared unfairly to methods that use additional 2D information, even the variant of SAT, which utilize real images during training and testing (row #2), we still outperform it by 1\%.
>
> | Method | Training       | Inference      |     Ref. Acc. |
> |--------|----------------|----------------|---------------|
> | SAT    | Real 2D Images |            X   |         47.3  |
> | SAT    | Real 2D Images | Real 2D Images |         47.9  |
> | LAR    |         SIG    |          SIG   |         48.9  |
> | LAR    | Real 2D Images | Real 2D Images |         46.8  |
>
> We argue that requiring extra 2D inputs limits potential application scenarios, such as sensor failure.
> Also, even with the existence of 2D information, it still contains a lot of drawbacks. For instance, synchronization between different sensors, such as in autonomous driving applications, is a costly operation.
> Hence, the 2D information could be distorted in certain conditions, such as low-light and challenging weather conditions.
>
> # ---------------------------------------------------------
> **9: The paper experiments show that it outperforms methods that have access to ground truth 2D images. How can this be explained? Are the single view images in the dataset taken from arbitrary angles? Is this paper just using a more advantageous camera angle? Or is this mainly due to the multi-view setting?**
>
> Generally, our SIG module is the essence of boosting the performance by leveraging the 2D information in a more 3D-aware way.
> Where the system acquires more robustness thanks to our multi-view setup and the camera augmentation technique.
>
> By tailoring a new variant of our architecture, as shown above, where our SIG module is replaced by the actual 2D clues produced by Faster R-CNN [4], we offer how significant our SIG module is.
> Where it outperforms SAT even when it utilizes the real 2D clues during the inference by 1\%.
>
>
>
> **References:**
>
> [1] Yang, Zhengyuan, et al. "Sat: 2d semantics assisted training for 3d visual grounding." Proceedings of the IEEE/CVF International Conference on Computer Vision. 2021.
>
> [2] https://github.com/CurryYuan/InstanceRefer/issues/4
>
> [3] https://github.com/zlccccc/3DVG-Transformer/issues/5
>
> [4] Salvador, Amaia, et al. "Faster r-cnn features for instance search." Proceedings of the IEEE conference on computer vision and pattern recognition workshops. 2016.
>
> [5] Referit3d: Neural listeners for fine-grained 3d object identification in real-world scenes.

---

> ### Author Response · Authors · 2022-08-08
> **Author response to Reviewer yWd9 [Part 4]**
>
> # ---------------------------------------------------------
> **10: The paper focuses on indoor scenes by how it selects the viewing directions of the virtual cameras. How would this work in more open-world settings where the camera does not always point outwards? Would this approach still work?**
>
> Our SIG module is generic enough to work for indoor and outdoor scenarios.
> For instance, in an application such as autonomous driving, a real camera or rig of cameras is mounted on the ego vehicle.
> Therefore, the position of the ego vehicle in the open-world space could be utilized while allocating our virtual cameras.
>
> To validate this, we applied our SIG module on one random sample from the KITTI-360 dataset. This figure (https://www.filemail.com/d/fyjluriehigbjaw) demonstrates the original RGB image and two synthetic images of two different objects generated using our SIG module.
>
> In addition, we define our virtual camera using the Unified Camera Model (UCM), which has five parameters $I = [\gamma_x, \gamma_y, c_x, c_y, \zeta]$
> Following the UCM, the projection is defined as follows:
>  $\pi (x_i^{pc},I) = \begin{bmatrix}\gamma_x  \frac{x}{\zeta d+z}   \\\\ \\gamma_y  \frac{y}{\zeta d+z} \end{bmatrix} +  \begin{bmatrix}c_x  \\\\ c_y  \end{bmatrix} ,$ which enables us to model fisheye cameras which is widely used in the autonomous driving applications due to its wide coverage region.
>
> # ---------------------------------------------------------
> **[Updated-Experiments] 5: The paper does not contain a qualitative comparison of the detection performance with state-of-the-art.**
>
> We experimented our LAR module with the detector-generated 3D proposals using VoteNet detector.
> To this end, we adapted the Scanrefer architecture as follows:
> 1) 2D Encoder: Integrating a 2D encoder right after the predicted box proposals, i.e., SIG module and Tiny-ConvNext in our case, and Faster R-CNN [1] in SAT case.
> 2) Fusion Transformer: In LAR case, we added our visual transformer only as we aim to emphasize the contribution of our SIG module only. While in SAT case, we convert their multi-modal transformer to vision transformer by excluding the language embedding from it.
> 3) The same 3D prediction modules and language modules are adopted from Scanrefer, i.e., P++ and Voting module, and GloVe and GRU, respectively.
>
> We train three models:
> 1) The best variant of Scanrefer.
> 2) SAT-Detection-based architecture.
> 3) LAR-Detection-based architecture.
>
> We first train the Scanrefer for 50 epochs then freeze their 3D prediction modules and language modules for two epochs to train the new blocks added in SAT and LAR detection based architectures. Then, we train the whole architecture for two more epochs. For more details, please refer to the "Detector-Generated 3D Proposal" section in Appendix A in the revised paper version, where we added a figure to demonstrate the adaptations done on the ScanRefer architecture.
>
> As shown in the below table, the tailored LAR-Detection-based architecture outperforms both the original ScanRefer and the SAT-Detection-based architectures.
> Also, the results show that SAT is more vulnerable to the distortions in the detected proposals.
>
> |   Method  |    Acc@0.25    |     Acc@0.5    |
> |-----------|----------------|----------------|
> | ScanRefer |     40.41      |      26.08     |
> |    SAT    |     40.92      |      26.25     |
> |    LAR    |     **42.14**      |      **26.96**     |

---

### Official Review · Reviewer_G6JP · 2022-07-15

**Rating:** 5
**Confidence:** 3
**Soundness:** 2 fair
**Presentation:** 3 good
**Contribution:** 3 good

**Summary:**

This paper presents LAR, a novel method for the task of 3D language-visual grounding. The key idea of LAR is to leverage synthetic 2D images as an additional modality of data to the downstream object reference task. LAR outperforms all baselines on the selected 3D grounding benchmarks that do not use additional training data.


**Questions:**

Same as 1, 2, 5 raised above in the weakness section


**Limitations:**

Same as 3 mentioned above in weaknesses. Additionally I would have liked to see if the setup is set closer with object detection: instead of assuming access to object proposal, predict objects directly given language description, since principally it seems language should also help bounding box proposal and localization. However, this is out of the scope of this paper and has not factored into my scoring.

**Strengths And Weaknesses:**

=== Strengths ===
1. The paper is well written and the presentation is clear.
2. Ablation studies on the design choices are mostly thorough.
3. Strong empirical performance over previous methods.

=== Weaknesses ===
1. The proposed SIG module uses heavy heuristics and hand-engineering. I am wondering if this can be learned – given 3D scene point clouds and a single object proposal bounding box, predict (multiple) camera extrinsics. It seems this can be learned end-to-end as the projection function seems also differentiable.
2. While LAR uses BERT embeddings, I am wondering if switching to a image-language embedding will improve performance given the task requires modeling language and visuals together in the first place.
3. I am not sure if learning simple 1D conv kernel weights (seems more like learned weighted average…) is a good practice to fuse multi-view images, since it ignores the context.
4. Typo in L181-182.
5. The current model seems to ignore interaction among objects and predicts reference for them individually. I am wondering whether modeling objects together might make a difference in the performance.

---

> ### Author Response · Authors · 2022-08-03
> **Author response to Reviewer G6JP [Part 1]**
>
> We thank you for your valuable and thoughtful feedback. We are encouraged that you found our approach effective, our experiments comprehensive, our ablation studies thorough, and our writing precise. Below, we address your concerns and questions and incorporate all the feedback.
>
> # ---------------------------------------------------------
> **1: The proposed SIG module uses heavy heuristics and hand-engineering. I am wondering if this can be learned.
> Given 3D scene point clouds and a single object proposal bounding box, predict (multiple) camera extrinsic. It seems this can be learned end-to-end as the projection function seems also differentiable.**
>
> Thank you for your suggestions. It took us significant efforts to design a camera module in an effective way that can be robust against many cases and generalize on the three datasets that we evaluated.
> We leverage this prior knowledge to fully utilize the limited available data, where the ScanNet dataset consists of almost 1500 indoor scenes only. Developing an AI system that learns what we learn by designing a learning system such as differentiable rendering systems might require much more data.
> However, we agree that it can be an interesting future direction, and it is indeed worth being explored.
> We will make our code publicly available and hope that this may encourage future work exploring such ideas and integrating them into our pipeline and beyond.
>
> # ---------------------------------------------------------
> **2: While LAR uses BERT embeddings, I am wondering if switching to an image-language embedding will improve performance given the task requires modeling language and visuals together in the first place.**
>
> Thank you for your suggestions. The language classifier is achieving 99\% accuracy, which measures the ability to know the target object given an input sentence. However, we agree with you that using image-language embedding, such as CLIP, may make our multi-modal transformer operate better on top of both modalities. As the language embedding's primary goal is not to determine the class type of the refereed object but to interact with the visual embedding through our multi-modal transformer. Thus, we acknowledge that this is an exciting ablation to be added and extended for the visual embedding alongside linguistic, e.g., BERT, embedding. Therefore, we will try to conduct an experiment to explore whether it will fulfill our hypothesis or not.
>
>
> # ---------------------------------------------------------
> **3: I am not sure if learning simple 1D Conv kernel weights is a good practice to fuse multi-view images, since it ignores the context. seems more like a learned weighted average**
>
> As shown in Table 1 in the supplementary materials, we have conducted a simple comparison between two fusion methods, i.e., the non-learnable addition operation and learnable 1-D Convolution. The simple learnable fusion scheme; 1D-Conv., performs better than the none learnable scheme; addition. Intuitively, more complex fusion techniques can be explored; however, we think using a simple fusion technique, such as 1-D Conv., as it showed promising results in similar applications such as channel attention modules [1].
>
> # ---------------------------------------------------------
> **4: The current model seems to ignore interaction among objects and predicts reference for them individually. I am wondering whether modeling objects together might make a difference in the performance.**
>
> Our model does not ignore interaction among objects, where we encode each object individually using different semantic information types, i.e., visual semantics (2D \& 3D), and $x_i^{geo} \in \mathbb{R}^{1\times 30}$. The geometry information represents the spatial location of each object $i$ in the scene and encodes the virtual camera parameters. Thus, it can be interpreted as a positional encoding for the vision transformer, where the first level of interaction among objects is captured.
> Then, the second level of interaction, which is with language, happens through the multi-modal transformer.
>
> # ---------------------------------------------------------
> **5: I would have liked to see if the setup is set closer with object detection: instead of assuming access to object proposal, predict objects directly given language description, since principally it seems language should also help bounding box proposal and localization.
> However, this is out of the scope of this paper and has not factored into my scoring.**
>
> Thanks for pointing out this direction. We are working on the detection-based experiment based on the ScanRefer dataset, where we don't assume access to the GT proposals, and once it is done, we will share the results with you.
>
>
> **References:**
>
> [1] ECA-Net: Efficient Channel Attention for Deep Convolutional Neural Networks. CVPR, 2021.

---

> ### Author Response · Authors · 2022-08-08
> **Author response to Reviewer G6JP [Part 2]**
>
> # ---------------------------------------------------------
> **[Updated-Experiments] 5: I would have liked to see if the setup is set closer with object detection: instead of assuming access to object proposal, predict objects directly given language description, since principally it seems language should also help bounding box proposal and localization.
> However, this is out of the scope of this paper and has not factored into my scoring.**
>
> Thank you very much for pointing this out.
> To validate our SIG module robustness with the predicted proposals, we experimented with the detector-generated 3D proposals using VoteNet detector.
> To this end, we adapted the Scanrefer architecture as follows:
> 1) 2D Encoder: Integrating a 2D encoder right after the predicted box proposals, i.e., SIG module and Tiny-ConvNext in our case, and Faster R-CNN [1] in SAT case.
> 2) Fusion Transformer: In LAR case, we added our visual transformer only as we aim to emphasize the contribution of our SIG module only. While in SAT case, we convert their multi-modal transformer to vision transformer by excluding the language embedding from it.
> 3) The same 3D prediction modules and language modules are adopted from Scanrefer, i.e., P++ and Voting module, and GloVe and GRU, respectively.
>
> We train three models:
> 1) The best variant of Scanrefer.
> 2) SAT-Detection-based architecture.
> 3) LAR-Detection-based architecture.
>
> We first train the Scanrefer for 50 epochs then freeze their 3D prediction modules and language modules for two epochs to train the new blocks added in SAT and LAR detection based architectures. Then, we train the whole architecture for two more epochs. For more details, please refer to the "Detector-Generated 3D Proposal" section in Appendix A in the revised paper version, where we added a figure to demonstrate the adaptations done on the ScanRefer architecture.
>
> As shown in the below table, the tailored LAR-Detection-based architecture outperforms both the original ScanRefer and the SAT-Detection-based architectures.
> Also, the results show that SAT is more vulnerable to the distortions in the detected proposals.
>
> |   Method  |    Acc@0.25    |     Acc@0.5    |
> |-----------|----------------|----------------|
> | ScanRefer |     40.41      |      26.08     |
> |    SAT    |     40.92      |      26.25     |
> |    LAR    |     **42.14**      |      **26.96**     |

---

### Official Review · Reviewer_J6Di · 2022-07-20

**Rating:** 5
**Confidence:** 4
**Soundness:** 3 good
**Presentation:** 2 fair
**Contribution:** 2 fair

**Summary:**

This paper proposes the SIG module to project RGB-D point clouds of object proposals to multiple views of RGB images, which acts as another stream of visual clues to help point cloud-based 3D visual grounding. In contrast to SAT, this method does not rely on actual 2D images to generate the 2D clues. The proposed method, also called LAR, fuses the aforementioned 2D visual tokens and 3D visual tokens by a visual transformer, and then the enhanced tokens in each stream are respectively fused with language tokens for the identification of referred objects, while these two streams of visual-linguistic tokens can be fused again by a fusion transformer for the final referring identification. The experiments on Nr3D, Sr3D, and Scanrefer datasets, show that the proposed SIG can consistently improve the performance in comparison to the SOTA.

**Questions:**

It is encouraging if the authors can answer the questions listed in the paper's weaknesses.

**Limitations:**

Yes, the authors have adequately addressed the limitations and potential negative societal impact of their work.

**Strengths And Weaknesses:**

--- Strengths ---

1. The proposed SIG module can directly exploit the 2D visual clues from RGB-D point clouds, without acquiring real 2D images associated with point clouds, thus is more flexible than SAT.
2. The experiments contain a list of comparisons, validating that the proposed SIG can consistently improve the performance than the way that SAT is used for exploiting 2D visual data.

--- Weaknesses ---

1. Other than the SIG module, the overall novelty of this paper is limited, in comparison to SAT. However, the proposed LAR model has to explicitly employ the SIG module both at the training and testing stages, it thus requires more computational budgets than the previous 2D-3D combined methods, such as SAT. The computational cost of the LAR model should be discussed.
2. The SIG module may require accurate object proposals for the rendering of 2D images, which is okay for Nr3D and Sr3D, but for actual scenarios such as ScanRefer, how to maintain the reliability of SIG-generated 2D data is not guaranteed. How will the performance degrade if the proposals are predicted?
3. Even with a bunch of sophisticated fusion techniques and extra computational cost introduced by the 2D visual stream, the proposed method just achieve marginal gains than SAT, which harms the significance of the claimed contributions.
4. The presentation of this paper needs significant improvement. For example,
    - Tab. 2: What do referit3D and non-SAT mean in this table?
    - Tab. 3: is SAT$^\dagger$ means the re-implemented SAT?
    - How the camera and image augmentations are applied in the experiments other than Tab. 4.
    - A number of typos and grammar errors throughout the whole paper.

---

> ### Author Response · Authors · 2022-08-03
> **Author response to Reviewer J6Di [Part 1]**
>
> We thank you for your valuable and thoughtful feedback.
> We are encouraged that you find our approach effective and well motivated and our experiments comprehensive with good results.
> Below, we will address your concerns and questions and incorporate all the feedback.
>
> # ---------------------------------------------------------
> **1: Other than the SIG module, the overall novelty of this paper is limited, in comparison to SAT.**
>
> We agree that there are some similarities to SAT on leveraging 2D and 3D clues to improve visual grounding, but there are also significant differences that we introduce:
>
> - Even with the existence of the real 2D images, we still surpass its performance by excluding the instant search retrieval system, such as Faster R-CNN [1], and by utilizing the multi-view capability, which is not an option while using actual 2D images. Furthermore, utilizing the multi-view capability makes our approach more 3D oriented as it incorporates more knowledge from different angles.
> To validate this, we tailored a variant of our architecture, where our SIG module is replaced by the actual 2D clues produced by Faster R-CNN [1].
> As shown in the table below, replacing our SIG module with the features produced by Faster R-CNN [1] degrades the performance by 2\%.
> In addition, even when we unfairly compared to methods that use additional 2D information, even the variant of SAT, which utilize real images during training and testing (row #2), we still outperform it by 1\%.
>
> | Method | Training       | Inference      |     Ref. Acc. |
> |--------|----------------|----------------|---------------|
> | SAT    | Real 2D Images |            X   |         47.3  |
> | SAT    | Real 2D Images | Real 2D Images |         47.9  |
> | LAR    |         SIG    |          SIG   |         48.9  |
> | LAR    | Real 2D Images | Real 2D Images |         46.8  |
>
> - SAT requires instant search retrieval system, which is trained offline, such as Faster R-CNN [1], to sample the most suitable 2D clues/learned representation from the object bounding boxes. In contrast, our learning system is End-to-End friendly over both 2D and 3D clues since the whole pipeline relies on the 3D input only and our 2D encoder is trained alongside the rest of the network.
>
> - More Efficient w.r.t the memory footprint and training time.
> Where the instant search retrieval system [1] utilized by SAT is pre-trained on the Visual Genome dataset [2], which even goes beyond ImageNet knowlege [3], which we use while pre-training our Tiny-ConvNext backbone [4].
>
> # ---------------------------------------------------------
> **2: The SIG module may require accurate object proposals for the rendering of 2D images, which is okay for Nr3D and Sr3D, but for actual scenarios such as ScanRefer, how to maintain the reliability of SIG-generated 2D data is not guaranteed. How will the performance degrade if the proposals are predicted?**
>
> Of course, the predicted proposals' quality considerably affects the referring accuracy; however, we argue that our approach is more robust than SAT against such distortions, thanks to accumulating the clues from multiples views. Our multi-view setup enables us to overcome the occlusion or the distortion caused by predicted proposals leveraging the other views while constructing our 2D grid.
>
> SAT requires an instant search retrieval system, such as Faster R-CNN [1], to sample the most suitable 2D clues/learned representation from a single view-based bounding box, which means that SAT performance will heavily degrade based on the quality of the proposals.
>
>
> **References:**
>
> [1] Salvador, Amaia, et al. "Faster r-cnn features for instance search." Proceedings of the IEEE conference on computer vision and pattern recognition workshops. 2016.
>
> [2] Krishna, Ranjay, et al. "Visual genome: Connecting language and vision using crowdsourced dense image annotations." International journal of computer vision 123.1 (2017): 32-73.
>
> [3] Deng, Jia, et al. "Imagenet: A large-scale hierarchical image database." 2009 IEEE conference on computer vision and pattern recognition. Ieee, 2009.
>
> [4] Liu, Zhuang, et al. "A convnet for the 2020s." Proceedings of the IEEE/CVF Conference on Computer Vision and Pattern Recognition. 2022.

---

> ### Author Response · Authors · 2022-08-03
> **Author response to Reviewer J6Di [Part 2]**
>
> **3: Even with a bunch of sophisticated fusion techniques and extra computational cost introduced by the 2D visual stream, the proposed method just achieve marginal gains than SAT, which harms the significance of the claimed contributions.**
>
> LAR outperforms all existing approaches that did not use additional information than the 3D point clouds by a significant margin, i.e., 5.0\% to the runner-up. Even when we unfairly compared methods that use additional 2D information, such as SAT, we still outperformed it by 1.6\%.  In addition, SAT introduces a huge overhead during the training due to leveraging an instant search retrieval system, such as Faster R-CNN [1], to sample the most suitable 2D clues/learned representation from the object bounding boxes. We believe that these results are significant.
>
> # ---------------------------------------------------------
> **4: Tab. 2: What do referit3D and non-SAT mean in this table?**
>
> Table 2 shows the ablation study on Nr3D of the 2D synthetic image resolution and multi-view setup. We rely only on the 2D stream, i.e., SIG, 2D encoder, 2D multi-modal transformer, and 2D head. In this setup, we adapt Referit3D and non-SAT variants by replacing the 3D stream, i.e., P++, with the aforementioned 2D stream.
>
> # ---------------------------------------------------------
> **5: Tab. 3: is SAT$\dagger$ means the re-implemented SAT?**
>
> The $\dagger$ symbol indicates we retrain the model for a fair comparison, as we, among others, faced difficulties reproducing SAT in the paper, despite our use of the original implementation. Others also face similar problems; see GitHub issues [5][6][7].
>
> # ---------------------------------------------------------
> **6: How the camera and image augmentations are applied in the experiments other than Tab. 4.**
>
> The best setup was followed in the rest of the tables, where both the camera and the image augmentations are utilized.
> Please refer to the "Camera vs. Image Augmentation" paragraph under Sec. 4.1 for the implementation details of both augmentation techniques.
>
> # ---------------------------------------------------------
> **7: A number of typos and grammar errors throughout the whole paper.**
>
> We thank you for carefully reviewing our paper and for your input. Please see the updated rebuttal version, where we incorporated the feedback.
>
>
>
>
> **References:**
>
> [1] Salvador, Amaia, et al. "Faster r-cnn features for instance search." Proceedings of the IEEE conference on computer vision and pattern recognition workshops. 2016.
>
> [2] Krishna, Ranjay, et al. "Visual genome: Connecting language and vision using crowdsourced dense image annotations." International journal of computer vision 123.1 (2017): 32-73.
>
> [3] Deng, Jia, et al. "Imagenet: A large-scale hierarchical image database." 2009 IEEE conference on computer vision and pattern recognition. Ieee, 2009.
>
> [4] Liu, Zhuang, et al. "A convnet for the 2020s." Proceedings of the IEEE/CVF Conference on Computer Vision and Pattern Recognition. 2022.
>
> [5] https://github.com/zyang-ur/SAT/issues/1
>
> [6] https://github.com/zyang-ur/SAT/issues/3
>
> [7] https://github.com/zyang-ur/SAT/issues/2

---

> ### Author Response · Authors · 2022-08-08
> **Author response to Reviewer J6Di [Part 3]**
>
> # ---------------------------------------------------------
> **[Updated-Experiments] 2: The SIG module may require accurate object proposals for the rendering of 2D images, which is okay for Nr3D and Sr3D, but for actual scenarios such as ScanRefer, how to maintain the reliability of SIG-generated 2D data is not guaranteed. How will the performance degrade if the proposals are predicted?**
>
> Of course, the predicted proposals' quality considerably affects the referring accuracy; however, we argue that our approach is more robust than SAT against such distortions.
> Thanks to accumulating the clues from more than one view.
> Our multi-view setup enables us to overcome the occlusion or the distortion caused by predicted proposals leveraging the other views while constructing our 2D grid.
>
> Hence, SAT requires an instant search retrieval system, such as Faster R-CNN [1], to sample the most suitable 2D clues/learned representation from a single view-based bounding box, which means that SAT performance will heavily degrade based on the quality of the proposals.
>
> To validate our SIG module robustness against the distortions that may caused by the predicted proposals, we experimented with the detector-generated 3D proposals using VoteNet detector.
> To this end, we adapted the Scanrefer architecture as follows:
> 1) 2D Encoder: Integrating a 2D encoder right after the predicted box proposals, i.e., SIG module and Tiny-ConvNext in our case, and Faster R-CNN [1] in SAT case.
> 2) Fusion Transformer: In LAR case, we added our visual transformer only as we aim to emphasize the contribution of our SIG module only. While in SAT case, we convert their multi-modal transformer to vision transformer by excluding the language embedding from it.
> 3) The same 3D prediction modules and language modules are adopted from Scanrefer, i.e., P++ and Voting module, and GloVe and GRU, respectively.
>
> We train three models:
> 1) The best variant of Scanrefer.
> 2) SAT-Detection-based architecture.
> 3) LAR-Detection-based architecture.
>
> We first train the Scanrefer for 50 epochs then freeze their 3D prediction modules and language modules for two epochs to train the new blocks added in SAT and LAR detection based architectures. Then, we train the whole architecture for two more epochs. For more details, please refer to the "Detector-Generated 3D Proposal" section in Appendix A in the revised paper version, where we added a figure to demonstrate the adaptations done on the ScanRefer architecture.
>
> As shown in the below table, the tailored LAR-Detection-based architecture outperforms both the original ScanRefer and the SAT-Detection-based architectures.
> Also, the results show that SAT is more vulnerable to the distortions in the detected proposals.
>
> |   Method  |    Acc@0.25    |     Acc@0.5    |
> |-----------|----------------|----------------|
> | ScanRefer |     40.41      |      26.08     |
> |    SAT    |     40.92      |      26.25     |
> |    LAR    |     **42.14**      |      **26.96**     |
>
> # ---------------------------------------------------------
> **8: LAR model has to explicitly employ the SIG module both at the training and testing stages, it thus requires more computational budgets than the previous 2D-3D combined methods, such as SAT. The computational cost of the LAR model should be discussed.**
>
> We compared our complexity against SAT and SAT`, during both the training and testing. SAT` is a variant of SAT where the additional 2D information are utilized during both training and testing.
> As shown in the below table, our training and inference time are 3.6 x and 0.7 x, respectively, compared to SAT.
> All the reported results are measured on single GTX-1080 GPU.
>
> | Method |    Ref. Acc.  |Trainable Parameters|Inference Parameters|Training Time|Inference Time|
> |--------|---------------|---------------|---------------|---------------|---------------|
> | SAT    |      47.3     |      237 M    |     **81 M**      |    0.317 FPS  |    **5.97 FPS**   |
> | SAT`    |      47.9     |      237 M    |     237 M     |    0.317 FPS  |    1.27 FPS   |
> | LAR    |      **48.9**     |      **118 M**    |     118 M     |    **1.152 FPS**  |    4.12 FPS   |
>
>
> [1] Salvador, Amaia, et al. "Faster r-cnn features for instance search." Proceedings of the IEEE conference on computer vision and pattern recognition workshops. 2016.

---

### Author Response · Authors · 2022-08-03
**LAR paper update**

We thank the reviewers for their valuable and thoughtful feedback.
We are encouraged that they find our approach effective, well motivated and novel ("G6JP", "yWd9", "8Se9"), our experiments comprehensive with good results ("J6Di", "G6JP", "yWd9", "8Se9"), our ablation studies are thorough which verify the effectiveness of designed parts ("J6Di", "G6JP", "yWd9", "8Se9"), and well written ("G6JP", "8Se9").

Key changes we made to the paper:

- As advised by Reviewers "J6Di", "G6JP", "yWd9", "8Se9", we experimented our LAR module with the detector-generated 3D proposals using VoteNet detector. For more details, please refer to the "Detector-Generated 3D Proposal" section in Appendix A in the revised paper version.

- As advised by Reviewer "J6Di", we compared our complexity against SAT, during both the training and testing. For more details, please refer to the "Computational Cost" section in Appendix A in the revised paper version.

- As noticed by Reviewer "8Se9", we corrected Figure 3 as it contains a minor mistake related to the 2D predicted labels Z`.

- As advised by Reviewer "yWd9", we shortened Section 3.1, and make it more clear by defining the camera axis, and the camera model that we used (UCM).

- As advised by Reviewer "yWd9", we added some clarifications to give the reader the direct intuition of why we formulate our 2D semantics in such way.

- Also as advised by Reviewer "yWd9" and "8Se9", we detailed the evaluation metrics in Section 4 to make the paper self-contained and clearer.

- Also as advised by Reviewer "yWd9" and "8Se9", we detailed the losses in Section 3.2 to make the paper self-contained and clearer.

- Also as advised by Reviewer "8Se9", we added an ablation study, Section 4.1, to emphasize the contribution of our SIG module, by replacing our SIG module by the real 2D clues obtained from the real 2D images, to mimic the 2D encoder used by SAT.

Below, we address the reviewer’s concerns individually and will incorporate all the feedback.

---

### Meta-Review · Area_Chair_SbkH · 2022-08-27

**Recommendation:** Accept
**Confidence:** Less certain

**Metareview:**

After the rebuttal and discussion the paper received one weak accept, and three borderline ratings (2 ba, 1br). The concerns of the only reviewer leaning towards rejection are well addressed, as such the AC sees no reason to reject this paper.

**Award:**

No

---

### Decision · Program_Chairs · 2022-09-14

Accept